# Cross-platform motif discovery and benchmarking to explore binding specificities of poorly studied human transcription factors

Ilya E. Vorontsov [1,28], Ivan Kozin [2,3,28], Sergey Abramov [1,4], Alexandr Boytsov[1,4], Arttu Jolma [5], Mihai Albu[5], Giovanna Ambrosini [6,7,8], Katerina Faltejskova [9,10], Antoni J. Gralak [11,12], Nikita Gryzunov[2,13], Sachi Inukai[14], Semyon Kolmykov[15], Pavel Kravchenko [16], Judith F. Kribelbauer-Swietek [11,12], Kaitlin U. Laverty [5], Vladimir Nozdrin [3], Zain M. Patel[5], Dmitry Penzar [1], Marie-Luise Plescher [17], Sara E. Pour[5], Rozita Razavi[5], Ally W. H. Yang[5], Ivan Yevshin[18], Arsenii Zinkevich [3], Matthew T. Weirauch [19], Philipp Bucher [12], Bart Deplancke [11,12], Oriol Fornes [20], Jan Grau [17], Ivo Grosse [17], Fedor A. Kolpakov[15,21], The Codebook/GRECO-BIT Consortium*, Vsevolod J. Makeev [1,22,27 ✉], Timothy R. Hughes [5 ✉] & Ivan V. Kulakovskiy [1,2,23 ✉]

A sequence motif representing the DNA-binding specificity of a transcription factor (TF) is commonly modelled with a positional weight matrix (PWM). Focusing on understudied human TFs, we processed results of 4,237 experiments for 394 TFs, assayed using five different experimental platforms. By human curation, we approved a subset of experiments that yielded consistent motifs across platforms and replicates, and evaluated quantitatively the cross-platform performance of PWMs obtained with ten motif discovery tools. Notably, nucleotide composition and information content are not correlated with motif performance and do not help in detecting underperformers, while motifs with low information content, in many cases, describe well the binding specificity assessed across different experimental platforms. By combining multiple PMWs into a random forest, we demonstrate the potential of accounting for multiple modes of TF binding. Finally, we present the Codebook Motif Explorer (https://mex.autosome.org), cataloguing motifs, benchmarking results, and the underlying experimental data.

Transcription factor (TF) binding to DNA is a crucial component of transcriptional regulation, responsible for coordinated gene expression within gene regulatory networks[1]. Alteration of TF-DNA interactions provides a major contribution to gene expression changes due to sequence variants[2]. Nucleotide sequences of DNA segments specifically recognized by a TF are usually in some sense alike, and are said to carry "a binding motif" or "motif occurrence". A rigorous definition of the motif is given in formal language theory[3] as a language, a special set of strings, the motif carriers, which is recognized by some finite automaton. In practice, "motif occurrences" are usually understood as sequence classification features, often localized in the sequence and readily visualized as a sequence logos[4]. Motifs are essential for annotating gene regulatory regions[5], interpreting regulatory variation[6], and

deciphering the logic learned by deep neural networks from genomics data[7]. Traditional and most popular representation of a motif is the position weight matrix (PWM, also called position-specific scoring matrix, PSSM). It has few independent parameters and assumes: independent contributions of neighboring nucleotides to the binding energy[8]. Despite the existence of many advanced motif models with interdependent nucleotide contributions[9], PWMs remain in the bioinformatics toolbox for studying gene regulation, and the databases providing TF binding motifs as PWMs, including CIS-BP[10], JASPAR[11], and HOCOMOCO[12], are actively used for predicting TF binding sites (TFBS).

A PWM is usually derived from a collection of related DNA sequences, for example, bound by a TF in some experiment. A plethora of experimental

A full list of affiliations appears at the end of the paper. *A list of authors and their affiliations appears at the end of the paper.
✉e-mail: seva.makeev@cruk.manchester.ac.uk; t.hughes@utoronto.ca; ivan.kulakovskiy@gmail.com

approaches have been developed to identify TFBS in random sequences, complete genomes, or their fragments[13]. However, isolating the contribution of the DNA sequence from features of the cellular or genomic contexts is challenging, especially when considering the data obtained in a living cell[14]. Data obtained in vitro avoids these issues and, with synthetic sequences, it is possible to explore the sequence space more uniformly. However, these methods also have their own technical biases, for instance, high-throughput SELEX (HT-SELEX)[15] saturates quickly with the strongest binding sequences[16]. Therefore, to overcome these challenges, the binding specificity of a TF ideally should be studied both in vivo and in vitro with both synthetic and genomic sequences, using multiple experimental platforms[17].

Until now, there have been very few systematic studies evaluating the performance of different motif discovery tools depending on the experimental assay supplying the data. The well-known large-scale benchmarking of motif discovery algorithms conducted by Tompa et al. in 2005[18] took place in the low-throughput era, and did not include any of the current experimental assays. A focused competition organized by Weirauch et al.[19] using in vitro protein-binding microarray (PBM) employed ChIP-seq as an external control but did not include motif discovery from experiments other than PBMs. Other studies either compared the performance of motif discovery tools only on simulated data[20] or evaluated only pre-existing PWMs[16,17,21].

Here we present the results of the Gene Regulation Consortium Benchmarking Initiative, GRECO-BIT, an offspring of the GRECO/GREEKC consortium[22] dedicated to building and benchmarking algorithms for DNA motif discovery and TFBS modeling. Here, in collaboration with Codebook, we performed a large-scale motif analysis of newly generated human TF binding data[23] obtained through five different experimental assays, using a variety of motif discovery tools followed by systematic benchmarking. Through comparative assessment of the resulting motifs, we developed the Codebook/GRECO-BIT Motif Explorer (Codebook MEX), an interactive catalog of motifs for 394 putative TFs that were analyzed in the Codebook dataset. This resource provides an overview of the efficacy of various tools for PWM-based motif discovery across different experimental platforms and highlights the PWMs with the highest overall rankings, thus laying the foundation for future benchmarking studies and paving the way for improved computational protocols for generating high-quality DNA sequence motifs.

## Results

In this study, we relied on the data from five experimental platforms used by the Codebook initiative[23] to assay the binding specificity of 394 proteins (see Workflow overview in Methods). Two platforms were used to delineate TFBS locations in the human genome: Chromatin immunoprecipitation followed by sequencing (ChIP-Seq[24]) and high-throughput SELEX with genomic DNA (GHT-SELEX[25]). The other three methods, standard high-throughput SELEX (HT-SELEX), selective microfluidics-based ligand enrichment followed by sequencing (SMiLE-Seq[26]), and protein binding microarray (PBM), were used to assess TF binding to synthetic DNA fragments with random sequences (e.g., 40 N random inserts for HT-SELEX or pseudo-random probes in the case of PBMs[27]). For HT- and GHT-SELEX, there were three variants differing in the protein production method: GST-tagged in vitro transcription (-IVT) with E. coli extracts, GFP-tagged IVT with wheat germ extracts (-GFPIVT), and whole human cell lysate (-Lys). The experiments covered many previously unexplored or incompletely profiled TFs, thus complementing existing databases, as well as a number of well-studied TFs (positive controls). To our knowledge, this is the first time that such a large collection of TFs has been assessed simultaneously in such a diverse set of experiments. This setup provides a unique opportunity for a cross-platform assessment of the performance of motif discovery tools with motifs derived from one experiment type tested with the data from other types of experiments.

For clarity, in this study, we use the term "motif" to refer to a localized DNA sequence pattern shared by DNA segments recognized by a TF, and "motif occurrences" or "motif hits" to denote individual pattern occurrences

(binding sites). A matrix of nucleotide frequencies summing to one in each matrix column (binding site position) is used for multiplicative scoring, and is called here as the Position Frequency Matrix, PFM. In turn, we use "PWM" to refer to a motif model in the form of a weight matrix used for additive scoring (usually containing log-odds values[8]). We adopted a systematic approach for motif discovery and benchmarking (Fig. 1A) starting with uniform preprocessing of the data, such as peak calling (for GHT-SELEX and ChIP-Seq data) and normalization (for PBMs), and splitting results of each experiment into training and test sets (see "Methods"). Dealing with poorly studied TFs, we did not expect each experiment to be technically successful, and each dataset to be usable for motif discovery or benchmarking. Thus, we conducted motif discovery in two rounds, focusing in the first round on assessing and curating experiments and in the second round on large-scale comparison of the motif discovery tools. In the first round of motif discovery, we applied nine software tools to the training data of all experiments. We used classic MEME[28] software, popular bioinformatics tools from the era of high-throughput data (HOMER[29], ChIPMunk[30], Autoseed[31], STREME[32], and Dimont[33]), and advanced methods (ExplaiNN[34] and, for selected datasets, RCade[35] and gkmSVM[36]). Not all tools were compatible with all data types, e.g., RCade was exclusively used for zinc finger TFs, and a specialized adaptation of Dimont for HT-SELEX (DimontHTS) was used for HT-SELEX data.

With the diverse set of platforms and different scenarios of motif usage employed, it is not trivial, if at all possible, to select a single universal benchmarking metric of motif performance. To evaluate the performance of all PWMs across the test data from all platforms, we employed multiple dockerized benchmarking protocols from Ambrosini et al.[17], with additions from Vorontsov et al.[12] and Kulakovskiy et al.[37], and adapted methodology for PBMs from Weirauch et al.[19] (see "Methods"). Technically, to scan a DNA sequence with a given motif, most of the employed benchmarking protocols use the sum-occupancy scoring[16]. Specifically for ChIP-Seq and GHT-SELEX peaks, we also ran the HOCOMOCO benchmark[37], which considers only a single top-scoring log-odds PWM hit in each sequence, and estimated the CentriMo motif centrality score, which accounts for the distance of the binding site to the peak summit[38]. For benchmarking, we converted all motif models to PFMs and PWMs.

Many of the TFs were previously uncharacterized, and the initial benchmarking therefore served as the criterion to determine which experiments were successful. To this end, initial benchmarking results underwent human expert curation to approve a subset of successful experiments for a detailed analysis. To approve the experiment, we required that either (1) motifs discovered from this experiment were consistently similar between platforms or similar to motifs for related known or Codebook TFs and scored highly in different benchmarks or (2) motifs discovered from and high-ranking on other approved experiments scored highly on the dataset of question. In other words, an experiment was approved if it directly yielded consistent motifs itself or if it provided high scores for consistent motifs from other experiments.

During curation (see "Methods") we took into account known motifs both to validate real cases (positive controls and Codebook TFs from well-studied families) and to exclude artifacts, or "passenger" motifs besetting multiple independent experiments. As a result, the approved set of experiments encompassed 236 TFs and comprised 1462 datasets. To expand the motif sets from popular tools (HOMER, MEME, RCade, STREME) and explore another advanced method (ProBound[39]), we additionally ran a second round of motif discovery. Of note, while ChIPMunk, Dimont, ProBound, and ExplaiNN were applied by their authors, other tools, including MEME and HOMER, were employed not from their creators' hands, and thus might be technically handicapped. Running the motif discovery in two rounds allowed to reduce the computational load as only approved experiments were explored in the second round.

In total, this effort generated 219,939 PWMs, with 164,570 derived from the approved experiments. Out of these, 159,063 PWMs passed additional automatic filtering for common artifact signals (such as simple repeats and the most widespread ChIP contaminants; see "Methods") and

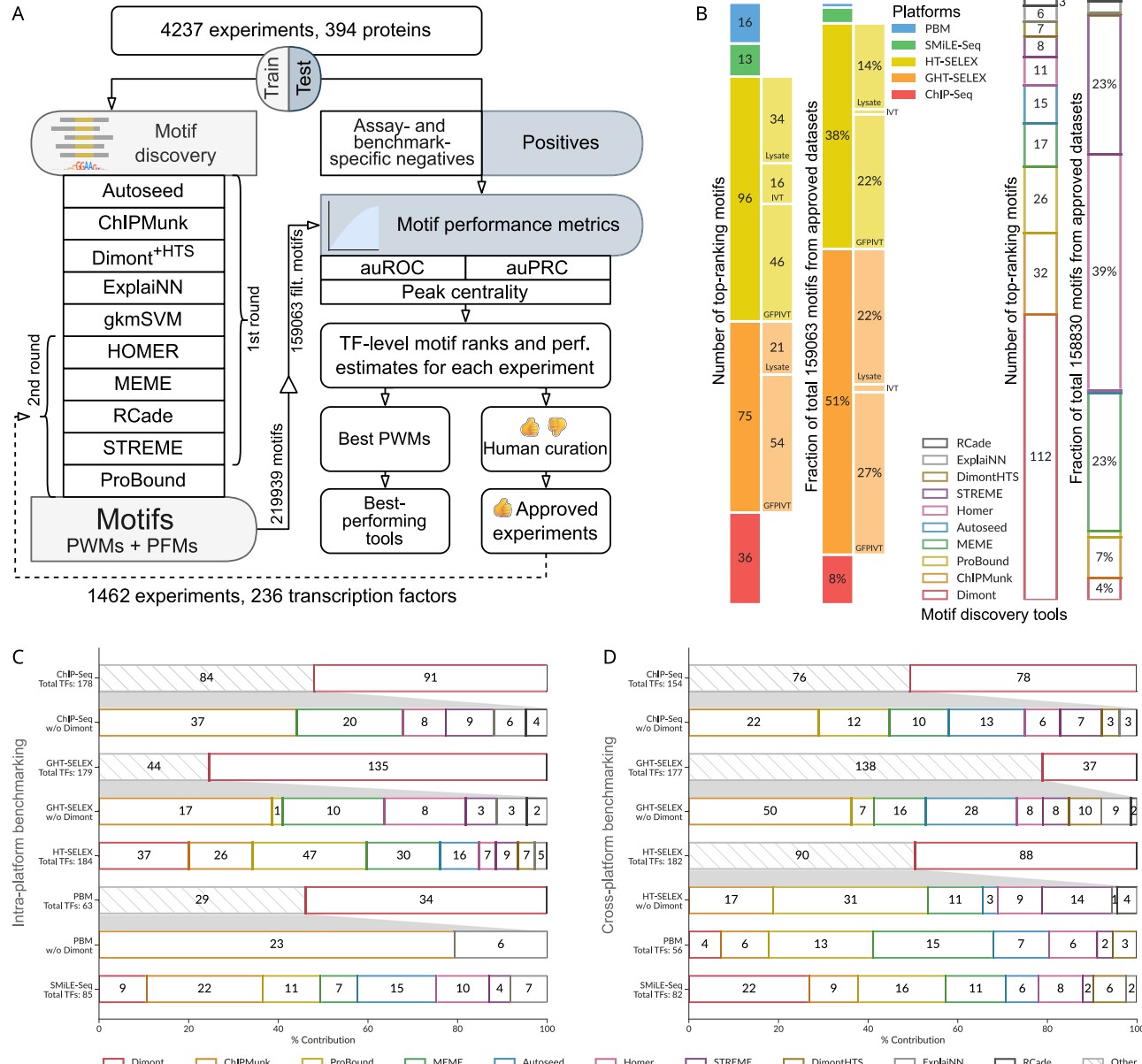

**Fig. 1 | Motif discovery and benchmarking pipeline, and the collection of top-ranking motifs. A** Schematic of the pipeline. **B** Contributions of different tools and experimental methods (types of GHT-SELEX and HT-SELEX are shown in extra bars) to the top-ranking motif collection (numbers of TFs) and to the complete MEX set of benchmarked motifs (expressed as a percentage). **C, D** Fraction from total (X axis) and absolute number of TFs with the top-ranking motifs produced by each motif discovery tool. The total number of eligible TFs (with the train, test data, and motifs from the same platform) is given in the vertical axes labels. For clarity, data for

tools other than Dimont are shown in nested callouts. **C** Intra-platform testing: the motifs are constructed from and tested on the same type of experiment. **D** Cross-platform testing, motifs obtained from datasets of all experimental platforms, excluding the one used for testing (labeled at the vertical axis). As in (**C**), only the top-ranking motifs for each TF are counted. TF transcription factor, PWM position weight matrix, PFM position frequency matrix, auROC area under the receiver operating characteristic, auPRC area under the precision-recall curve.

formed the primary motif set for the downstream analysis (Supplementary Fig. 1A, B). The resulting data are available at ZENODO[40–42], including the train-test sets and 16,869,771 performance estimates for all <motif, dataset> pairs for each TF. 9,339,595 of those belong to the motifs from approved experiments. Interactive access to both the approved and complete motif collections, alongside the benchmarking results, is provided by the Code-book Motif Explorer (Codebook MEX, https://mex.autosome.org).

### Benchmarking reveals versatility of individual approaches and added value of multiple tools

Using the data from approved experiments, we sought to construct a global benchmarking ranking for different motif discovery algorithms across

different experimental platforms. To this end, we employed a hierarchical ranking procedure to sequentially identify the top-ranking motifs for each TF and experiment type across multiple individual performance metrics, replicates, and types of experiments, followed by a global TF-level ranking (see "Methods").

We expected that the benchmarking study would reveal a motif discovery tool to be either universally superior or best suited to derive motifs from particular types of experiments. However, these expectations were only partially met. On the one hand, nearly all software tools contributed to the final collection of globally top-ranking motifs per TF (Fig. 1B), i.e., there was no algorithm, in retrospect, that was not worth including (the one possible exception was gkmSVM, which, due to high computational load,

was applied only to a subset of ChIP-Seq data, resulting in the smallest initial set of motifs, and was tested here using derived PWMs, rather than its native sequence scanner). On the other hand, nearly half of the top-ranking motifs in the final collection were generated by a single tool, Dimont. This dominance persisted even when considering the top 20 motifs for each TF (Supplementary Fig. 1C). Furthermore, the proportional contribution of the tools to the collection of top-ranking motifs did not reflect the initial quantity of motifs generated by these tools (Fig. 1B). In contrast, the distribution of experiment types that yielded the top-ranking motifs was more similar to the original composition of the Codebook data: the top-ranking motifs were largely derived from ChIP-Seq and (G)HT-SELEX experiments (Supplementary Data 1).

When considering individual experiment types, we first examined which tool produced the top-ranking motifs across TFs when trained and tested on data from the same type of experiment (Fig. 1C, Supplementary Fig. 2). Dimont was the top performer in three categories (GHT-SELEX, ChIP-Seq, PBM), and was competitive for HT-SELEX, which constituted the majority of the data, thus explaining its significant contribution to the global set of top-scoring motifs. ProBound led in HT-SELEX, while ChIP-Munk was the second best for many types of experiments, and led for SMiLE-Seq. This comparison highlights tools that excel at capturing experiment-specific motifs, potentially including biases and artifacts in addition to the intrinsic binding specificity of the TF.

Next, we conducted a cross-platform analysis, where we assessed TFBS prediction performance for a particular type of experiment using motifs discovered from all other types (Fig. 1D, Supplementary Fig. 2). Considering top-ranking motifs, Dimont scored highest overall for SMiLE-Seq, HT-SELEX, and ChIP-Seq. Conversely, ChIPMunk excelled with genomic HT-SELEX and MEME with PBMs. Interestingly, ProBound was powerful in predicting PBM and SMiLE-Seq, and to a lesser extent, HT-SELEX, suggesting its ability to capture lower-affinity binding sites common in PBM and SMiLE-Seq data. Analyzing the variants of GHT-SELEX and HT-SELEX individually (Supplementary Fig. 3), the results were similar (Dimont led in GHT-SELEX and ProBound in HT-SELEX), and the observed variability between experiment subtypes likely reflects the differences in the profiled TFs and the signal-to-noise ratios of particular experiments.

Summing up, considering overall motif rankings, Dimont excels across the board and particularly at ChIP-Seq, but intra- and cross-platform benchmarking highlight alternative motif discovery tools best suited for TFBS recognition in specific scenarios. Thus, on the one hand, less popular but powerful tools such as Dimont or ChIPMunk are worthy candidates for wider practical usage with the data from different platforms. On the other hand, tools excelling at particular platforms in general (e.g., ProBound for PBMs) or only for particular datasets should not be forgotten, at least as a complementary addition. For example, Autoseed successfully detected appropriate motifs in particularly noisy HT-SELEX datasets, while not being the best on average.

## The first motif reported is the best in benchmarking in 75% of cases

A single run of a particular motif discovery tool, be it MEME or Dimont, may yield multiple motifs, and ideally, it should put the true binding motif at the top of the list. In some datasets, however, the first reported motif may reflect the binding patterns of a TF cofactor (e.g., in ChIP-Seq) or even spurious signals such as artificially enriched sequences (e.g., aptamers in HT-SELEX). Yet, these examples are usually experiment-specific and unlikely to rank highly in the overall benchmarking across different platforms. Thus, motifs ranked higher in the overall benchmarking should have a higher probability of reflecting the true binding specificity.

For each run of a motif discovery tool on a particular dataset for a particular TF, we took the first three reported motifs (excluding common artifact signals, see "Methods") and located them in the overall benchmarking ranking (which includes motifs from all programs and all datasets for the TF). In about 75% of cases (runs of a particular software using a

particular training dataset that yielded more than one motif), in the global benchmarking, the first reported motif indeed scored higher than the other two (Supplementary Fig. 4A). However, in the remaining 25% of cases, the highest-ranked motifs from a particular motif discovery run were not the first in the software output, i.e., the internal ranking of the motif discovery tools failed to distinguish what we assume are the proper signals. In real-life scenarios, this percentage could be even higher as a priori pre-filtering of common artifacts would not be possible without multi-platform data. Therefore, in practice, secondary motifs reported by motif discovery tools must be considered in downstream analyses. Of note, Dimont and ChIP-Munk stood out in this test, and for these tools, the first reported motif was relevant in 90–95% of cases.

## Quantitative analysis of motif performance

While ranking analysis provides a bird's-eye view, it does not reveal the actual performance difference between the winning and runner-up motifs. Yet, direct comparison across TFs and datasets is complicated as the effective range of the achievable performance metric values depends not only on the benchmarking protocol, but also on the TF and experimental platform, and often differs between replicates. Thus, to compare the motifs and motif discovery tools quantitatively, we introduced an overall "harmonized" metric of motif performance across different benchmarks and datasets yielded by a particular experimental platform. Briefly (see "Methods" for more details), for each quadruple of a TF, a dataset, a benchmark (e.g., binary classification of bound and non-bound sequences), and a performance metric (e.g., auROC) we rescaled the values across motifs into the range [0,1], 0 corresponding to the worst and 1 to the best of values achieved by different motifs. The overall performance of a motif is the average of those rescaled values across all metrics and experiments of a particular type. For each TF and each motif discovery tool, we then selected representative motifs achieving the highest overall performance in the intra- and cross-platform analysis with particular target experiment types.

Figure 2 displays the median and the interquartile range (IQR) of the overall performance of representative motifs across TFs assayed in the intra- and cross-platform fashion (Fig. 2A, B). In these tests, we expected the tools with wider applicability across TFs to yield a higher median and lower IQR. In the intra-platform comparison, HT-SELEX appeared to be the most agnostic to the motif discovery tool, as the median overall performance was consistently high with low IQR across TFs for all tools. In contrast, the largest differences between tools were observed for SMiLE-Seq data. From the tool-centric perspective, the mean performance of Dimont, ChIPMunk, and Autoseed was stable across TFs and platforms, although MEME and HOMER were not far behind. RCade performance on SMiLE-Seq data is noteworthy, as it was applied only to a small subset of TFs, but the resulting motifs displayed a strong performance. Finally, ExplaiNN could not quantitatively compete with other tools when trained and validated on genomic regions, potentially due to a lower-than-necessary volume of available training data. In the cross-platform setting, the differences between the tools lessened: all conventional tools (MEME, STREME, HOMER, ChIPMunk, Dimont) performed comparably well. We rationalize this outcome by the fact that the representative motifs were collected across platforms, allowing each tool to avoid individual within-platform pitfalls, although gkmSVM displayed acceptable median results despite being trained on ChIP-Seq data only. Still, the IQR was high for many specific combinations of tools and experiment types, i.e., depending on a TF, any single tool can fail to properly capture a universally applicable motif even with multiple attempts across platforms, despite performing satisfactorily on average.

The scaling used to obtain the overall motif performance estimates conceals the information on the absolute efficacy of a tool. Indeed, even if all the tested motifs were of very high quality and achieved the auROC over 0.9 on a particular dataset, the respective scaled values were still stretched across the [0,1] band. To obtain more interpretable performance estimates, we plotted the distribution of raw values of the area under the receiver operating characteristic (auROC, computed with sum-occupancy PFM scoring as in

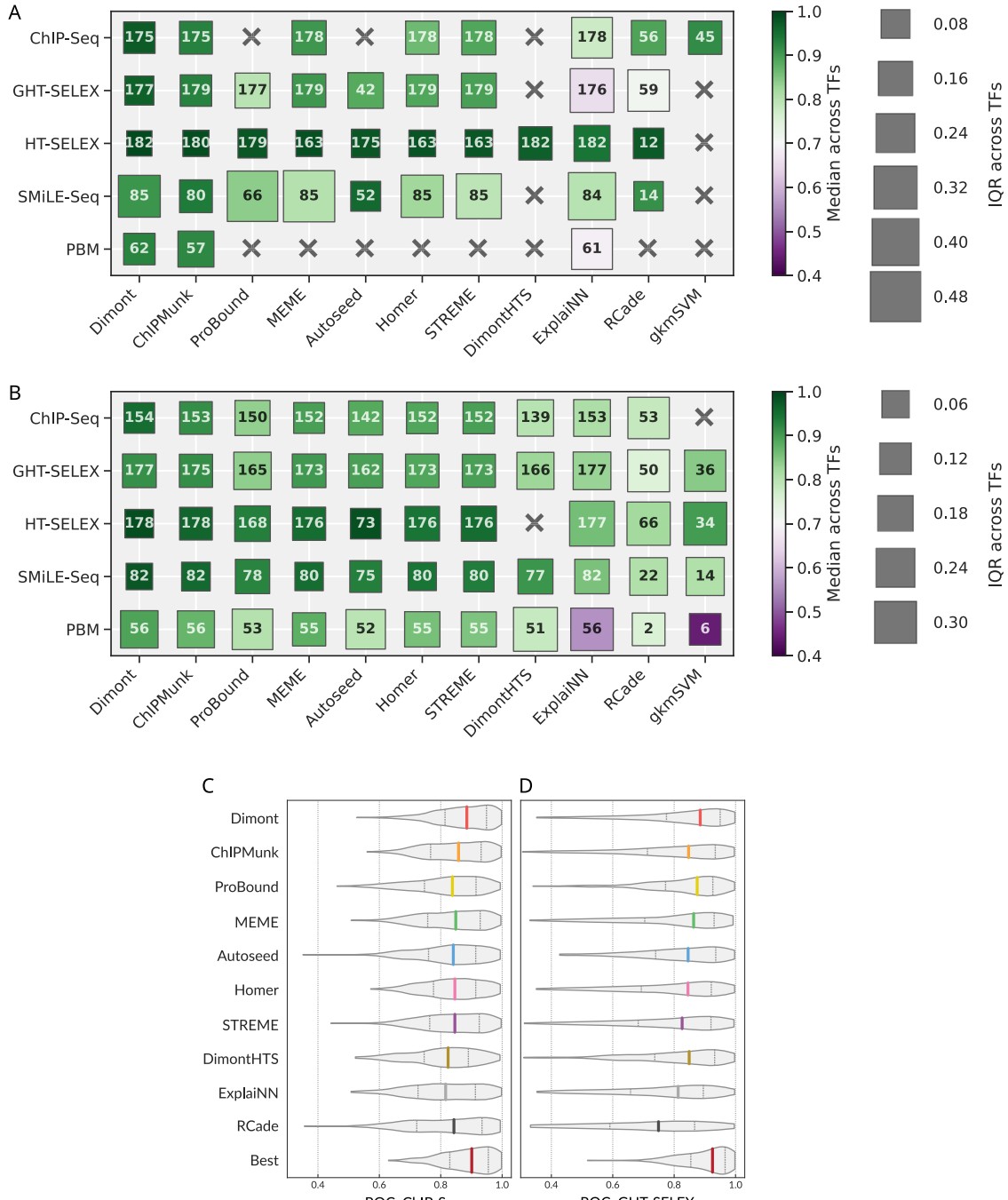

**Fig. 2 | Quantitative analysis of the intra- and cross-platform performance of different motif discovery tools. A** The highest overall performance of the best motifs (one per TF) when training and testing on the same type of experiment. **B** The highest overall performance of the best motifs (one per TF) in cross-platform evaluation. The color scale (identical in (**A**) and (**B**)) represents the median performance (the higher the better), and the size of the boxes (note the different scale between (**A**) and (**B**)) indicates the IQR (the lower the better) across TFs. The number in each square shows the total number of tested TFs for each combination of a motif discovery tool and an experiment type. **C, D** Distributions of auROC values for all TF-dataset pairs calculated from the top-ranking motifs from each motif discovery tool selected by global benchmarking: tested on ChIP-Seq (**C**), tested on all variants of genomic HT-SELEX (**D**). The bottom violin is built from the highest values obtained for different TF-dataset pairs, considering the top-ranking motifs from all tools. Colored dashes: median values, dotted dashes: the 1st and the 4th quartiles. auROC area under the receiver operating characteristic, IQR interquartile range, TF transcription factor.

Ambrosini et al.[17]) using top-ranking motifs from each tool. Violin plots illustrate the distribution of raw auROC values across TFs and test datasets for ChIP-Seq (Fig. 2C) or GHT-SELEX (Fig. 2D) data. The values reached by Dimont are consistently higher than those of other tools and close to those of the best PWMs (selected irrespective of the tool), with the median auROC across TFs and datasets over 0.85 for both ChIP-Seq and GHT-SELEX. A different performance metric, asymptotic pseudo-auROC

(computed with best log-odds PWM best hits as in Kulakovskiy et al.[37]), was less discriminative between tools (see "Methods" and Supplementary Fig. 4B).

The diversity of experimental platforms allows for answering another important question: whether protein binding experiments with synthetic oligonucleotides provide motifs suitable for the reliable prediction of genomic binding sites. For different TFs and different datasets, we plotted

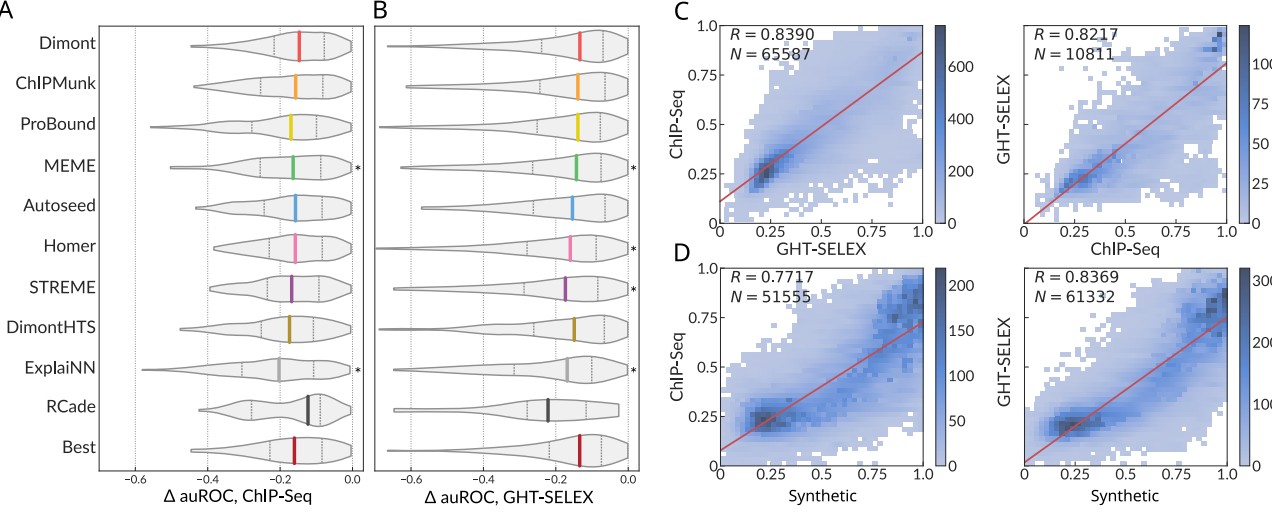

Differences between the best auROC values achieved by motifs derived from genomic and artificial sequences

Mean overall motif performance measured using training (X-axis) and different test (Y-axis) data types

**Fig. 3 | Overall performance of motifs derived from synthetic or genomic sequences when applied to prediction of genomic binding sites. A, B** The difference between the best auROC values for genomic data achieved by motifs discovered in the genomic (ChIP-Seq or GHT-SELEX) or synthetic (HT-SELEX, SMiLE-Seq, PBM) sequences; ChIP-Seq (**A**), GHT-SELEX (**B**). The last violin at the bottom shows the difference between the highest best-achieved values for each TF across all tools. $*p < 0.05$, one-tailed paired Wilcoxon test against the best-achieved

values (the bottom violin). IQR: interquartile range. **C, D** Comparison of the overall motif performance at the training data type (ChIP-Seq, GHT-SELEX, or any platform with synthetic sequences, X axes) and the test data (ChIP-Seq or GHT-SELEX, Y axes). Color scale: number of motifs; the total number of tested motifs (N) and Pearson's correlation (R) are labeled on the plot; a linear regression trend is shown as a red line.

the distributions of the difference between the maximal auROC values achieved with the genomic data (ChIP-Seq and GHT-SELEX) by motifs obtained from the genomic and the synthetic data (HT-SELEX, SMiLE-Seq, PBM), Fig. 3A, B. Overall, we observed a visible drop of auROC median of −0.1 to −0.2 depending on the tool, meaning that the genomic binding sites remain difficult to predict with motifs from synthetic data even when using multiple motif discovery tools. In extreme cases, for some datasets, the auROC dropped extremely low (ΔauROC < −0.5). However, there were many cases with only a marginal decrease of auROC as the area around ΔauROC near 0.0 is densely populated for many motif discovery tools.

A thorough analysis of the overall performance of motifs for all pairs of platforms (Supplementary Fig. 5) reveals that there are subsets of motifs achieving very high scores at the training and the test experiment types simultaneously for almost any train-test combination of platforms. As expected, the transfer between genomic platforms (ChIP-Seq to GHT-SELEX and vice versa) is highly reliable, and a better performance at the training data translates to a better performance at the test data (Fig. 3C). However, the transfer from synthetic to genomic data is complicated (Fig. 3D): while the performance estimates are also strongly correlated, low performance at the test (genomic, Y axes) data type sometimes is observed for satisfactory scores at the train data, note the dense cloud of motifs sticking to the X axes. Thus, only the top-scoring "synthetic" motifs reliably predict genomic binding sites and provide a good and generalized representation of the true binding specificity of a TF of interest. We caution, however, that multiple outliers are found in the whole range of performance scores; thus, high scores for a single training data type do not guarantee universal cross-platform transferability.

Considering individual tools, their performance generally followed the global trend, except for an unexpectedly good RCade performance when tested on ChIP-Seq but not GHT-SELEX data (Fig. 2C, Fig. 3A versus Fig. 2D, Fig. 3B). This effect permits a simple explanation: by design, RCade obtained motifs for zinc-finger TFs only. For these TFs, the performance ratings at the respective genomic datasets were higher for many other tools (Supplementary Fig. 4C), and binding sites in ChIP-Seq datasets were easier to predict than those in GHT-SELEX (Supplementary Fig. 4D).

**Interpreting the role of flanking regions with motifs from gkmSVM.** GkmSVM was used in the first round of motif discovery from ChIP-Seq data, covering 45 TFs with approved experiments. As gkmSVM was computationally demanding and its motifs were not top-ranking, we did not apply it to the analysis of the entire Codebook collection. However, gkmSVM motifs trained on the ChIP-Seq data performed competitively (Fig. 2A). By examining the motifs of the 45 TFs constructed with GkmExplain from gkmSVM results, we found that it captured long sequence contexts in the vicinity of the binding sites, as seen from the motif length distribution (Supplementary Fig. 6A). Given the good performance of gkmSVM on ChIP-Seq and its acceptable performance for GHT-SELEX (Fig. 2B), we concluded that the extended genomic context provided added value, at least for some TFs. Yet, the longer genomic context of binding sites may represent properties of regulatory regions at a larger scale, including binding sites of other interacting TFs, rather than the genuine binding specificity of the protein under study.

**Basic motif features are irrelevant for benchmarking performance.** Basic motif features like the motif length, information content (IC), and GC composition were irregular even for motifs derived with the same tool or experiment type. For instance, Dimont and ProBound produced many low-information content motifs, and for many tools, there were discrete spikes in preferred motif lengths arising from technical parameters or other technicalities of the motif discovery procedure (Supplementary Fig. 6A, B). The performance metrics were only weakly correlated to these basic motif features, as previously observed in Ambrosini et al.[17]. Considering genomic data, performance metrics were not correlated with GC composition or positional IC but did show a weak correlation with the motif length (Pearson ρ of 0.05 to 0.2, Supplementary Fig. 7A). This finding could reflect an ability of longer motifs to partially account for the contribution of the genomic context, as motif length was especially beneficial in the benchmark that used only the single best PWM hit per sequence (pseudo-auROC). In this scenario, an extra motif length optimization step could improve the resulting PWM performance for some tools[43], although there is some risk that the motif does not represent

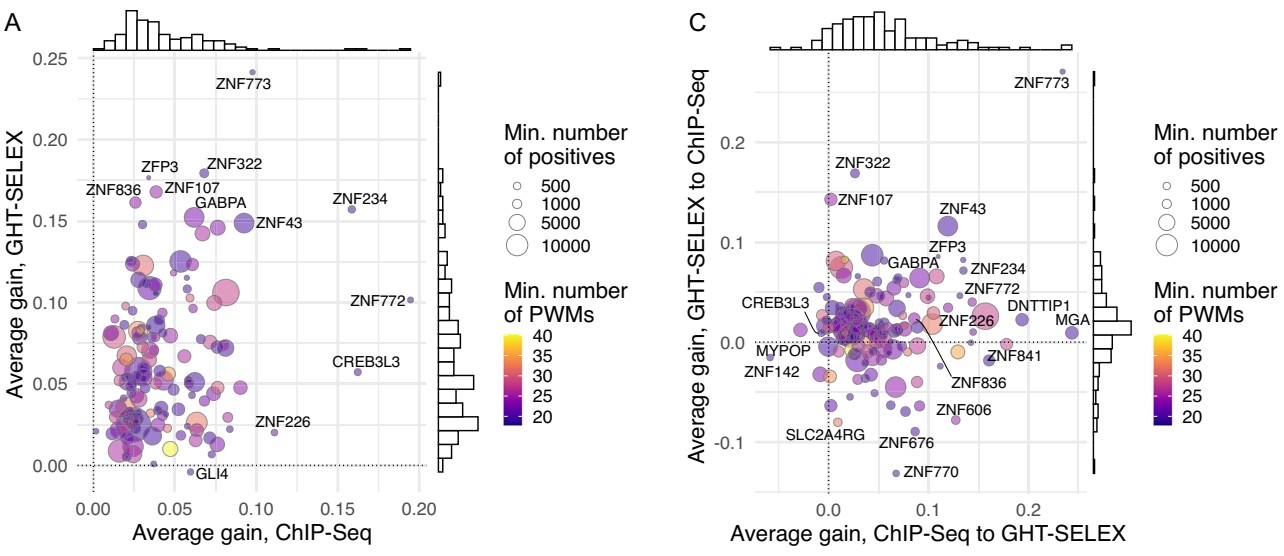

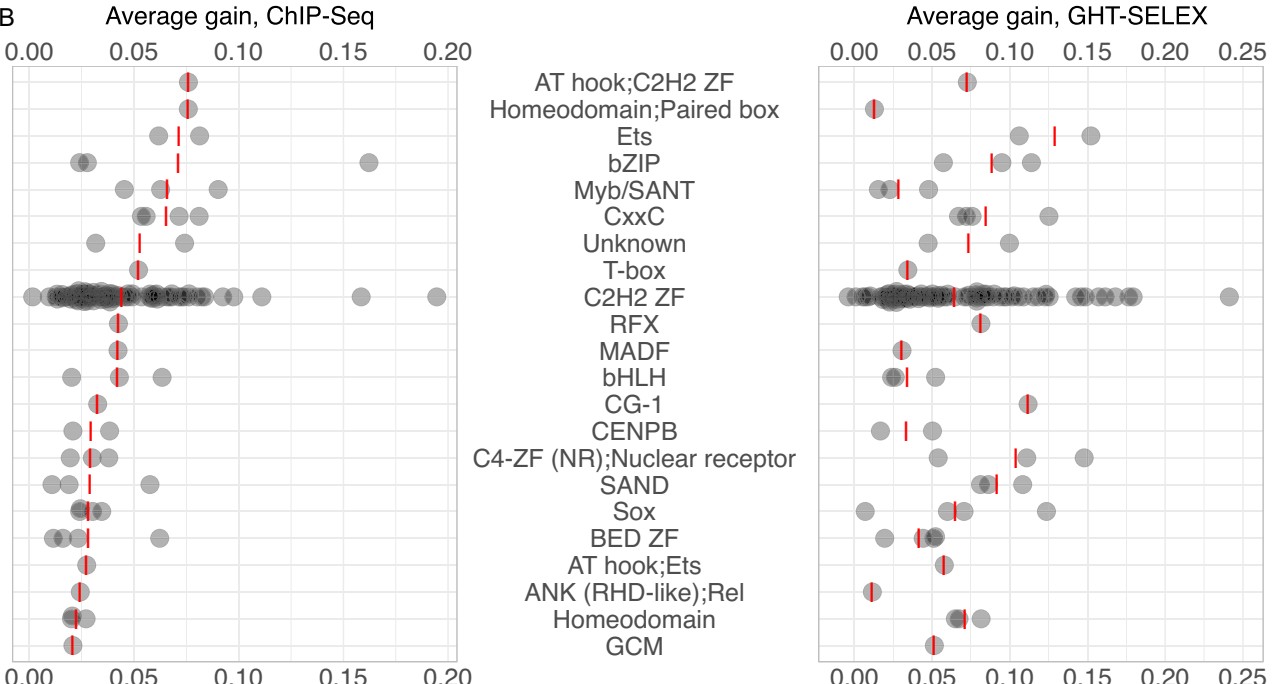

**Fig. 4 | Improved prediction of binding sites with a random forest of alternative motifs (Archipelago). A** Average gain (ΔauPRC and ΔauROC averaged) between the best PWM and Archipelago. X-axis: training and test with ChIP-Seq, Y-axis: training and test with GHT-SELEX, color scale: the number of PWMs included in the model (minimum of GHT-SELEX and ChIP-Seq), point size: the size of the training positive set (minimum of GHT-SELEX and ChIP-Seq). **B** Average gain

achieved for different TF families. Red dashes denote mean values, families are sorted by the median gain for ChIP-Seq. **C** Average of ΔauPRC and ΔauROC between the best PWM and Archipelago when training and testing on different experiment types. X-axis: transfer from ChIP-Seq to GHT-SELEX, Y-axis: transfer from GHT-SELEX to ChIP-Seq.

the intrinsic activity of the TF: in benchmarks on synthetic sequences, there was no correlation between motif length and performance at all, i.e., longer motifs did not have any advantages (Supplementary Fig. 7B) and only a very weak correlation (ρ around 0.06) with the positional IC. Focusing on the latter, a large IC spread was found even considering only the top 10 motifs for each TF (Supplementary Fig. 7C). Moreover, in our cross-platform assessment, the information content was not related to motif performance and instead reflected the motif origin experiment and motif discovery algorithm (Supplementary Fig. 6), suggesting that the average information content of a given high-scoring motif depends not

only on the experimental platform and specifics of a particular experiment (e.g., sequencing depth, TF concentration, or signal-to-noise ratio) but also on the technical procedure employed for motif discovery. The biophysical explanation for the success of many low IC motifs is uncertain, but one hypothesis that has been put forward is that they average multiple binding modes[44].

All in all, while it is desirable to know easy-to-compute motif properties to assess the motif reliability without any complex benchmarking (which requires both computational effort and extra experimental data), our large-scale benchmarking failed to find the necessary evidence, in agreement with

our previous benchmarking study with publicly available PWMs for well-studied TFs (see Figure S6 in Ambrosini et al.[17]).

## A random forest of PWMs improves prediction of genomic binding sites

Many TFs display several modes of DNA recognition with clearly distinguishable motif subtypes[12,13], and a straightforward advanced model could account for alternative TF binding modes by applying a logistic regression[45] or decision trees[46] on top of predictions from a collection of related PWMs. The richness of data provided by Codebook allows for a deeper exploration of such strategies.

Here, we exploited the high similarity of GHT-SELEX and ChIP-Seq data and considered the genomic TFBS prediction task for 137 TFs for which both GHT-SELEX and ChIP-Seq experiments were approved and generated enough peaks for analysis (see "Methods"). As evaluating advanced models is more sensitive to biases in the data, we created dedicated train-test datasets with several negative controls, including "shades" (i.e., peak-neighboring regions as in the main PWM benchmarking), "alien" peaks of non-relevant TFs, and "random" genomic regions. The latter two background sets were built by sampling from available sequences to achieve the same distribution of the GC composition as that of the positive set. Using the random negative set, we trained the *Archipelago* model (see "Methods"), a random forest classifier built on top of the best hits of a relatively small collection of log-odds PWMs (excluding those discovered from the test data type). Next, we estimated its mean performance with the three alternative negative sets and used PWMs reaching either the highest auROC or auPRC as the respective baselines.

**Intra-platform evaluation**. First, we evaluated Archipelago models using data solely from ChIP-Seq or GHT-SELEX. To avoid information leakage, each time, we excluded PWMs obtained from the test experiment type. Yet, it was still possible to train and test Archipelago using PWMs derived from other platforms. In this scenario, Archipelago consistently outperformed individual PWMs, with significant gains in both auROC and auPRC across TFs (Supplementary Fig. 8A–F, with the average of ΔauROC and ΔauPRC shown in Fig. 4A), although the random forest showed only a marginal improvement over logistic regression (Supplementary Fig. 8G). In agreement with the primary PWM benchmark, the absolute intra-platform auROC values were high even for individual PWMs (Supplementary Fig. 8A, D), while achieving high auPRC values was more difficult due to class imbalance. As expected, the negative set made of shades, which had a less skewed class imbalance, yielded the highest auPRC scores (Supplementary Fig. 8B, E). Overall, comparing different TFs, the improvement of Archipelago over PWMs did not depend on the size of the positive set or the number of PWMs in the model. Remarkably, only 2–4 PWMs combined were already sufficient for a major quality boost over a single PWM (Supplementary Fig. 8H, I).

The average performance gain for GHT-SELEX and ChIP-Seq differed for individual TFs and TF families (Fig. 4B). Some TFs, such as GABPA, which is known to form multimers on DNA[47], showed improvement with Archipelago presumably due to multiple distinct binding sites per peak not being captured by a single best PWM hit. Other TFs received the improvement due to the complexity of the binding patterns (Supplementary Fig. 9), which are hard to represent by a single fixed-width PWM. For example, it appears that the CREB3L3 ChIP-Seq fortuitously captured heteromeric binding[48]. Some TFs bind several site subtypes, such as C2H2 zinc finger TFs with modular binding specificities[25]: Archipelago improvement for ZNF772 and ZNF773 seems to be driven by their distinct motif subtypes, although for ZNF772 ChIP-Seq, Archipelago also relied on a low-complexity polyA motif. A strong example is ZNF43 with PWMs representing single and double-box binding motifs, which are conveniently taken into account together by the random forest but not by single PWMs (Supplementary Fig. 9).

**Evaluating the models' cross-platform transferability**. At first glance, the cross-platform performance of Archipelago (training the random forest on the ChIP-Seq and testing on GHT-SELEX and vice versa) seems

contradictory: ChIP-Seq-to-GHT-SELEX train-test yields a stable performance increase, while the train-test in the opposite direction behaves more randomly and often underperforms even in comparison to the best single PWM (Supplementary Fig. 10A–H, Fig. 4C, Supplementary Data 2). This can be explained by the data volume available for training from ChIP-Seq and GHT-SELEX, with the latter generally providing 2–3 times fewer peaks (Supplementary Fig. 10I). This difference not only made the model training more prone to overfitting but also provided less information on the actual diversity of genomic binding sites. Another issue arises from different motif subtypes, which are preferably represented in ChIP-Seq and GHT-SELEX peaks. A controversial example is FOSL2, for which the alternative motif subtypes were exclusively prioritized either in ChIP-Seq or in GHT-SELEX (Supplementary Fig. 9), but the cross-platform performance of Archipelago was still improved over a single PWM. Similarly, for ZNF770, a longer motif variant likely coming from genomic repeats was prioritized in ChIP-Seq but not in GHT-SELEX (Supplementary Fig. 9). Yet, in the end, the top TFs receiving the highest performance gain in the cross-platform evaluation were partly shared with those with the highest performance gain in the intra-platform setup, such as ZNF773, ZNF43, and GABPA.

## Codebook Motif Explorer

The results of this study are presented through the interactive Codebook/GRECO-BIT Motif Explorer (MEX, https://mex.autosome.org), which provides motifs, performance metrics, ranks, logos, and structured downloads, such as sets of top-performing motifs and related metadata, an overview is given in Supplementary Fig. 11. The complete set of MEX motifs and the benchmarking-ready Codebook data are also available at ZENODO[40–42].

## Discussion

Computational methods for motif discovery in DNA sequences have been evolving for more than three decades, stimulated by progress in experimental methods for profiling DNA-protein interactions. Yet, quantitative assessment of the performance and reliability of motif discovery tools is lagging, partly due to a shortage of uniform datasets for validation between experimental platforms. Thus, forty years after the inception of PWMs and following many advances in the measurement and representation of DNA sequence specificities, it remains controversial how to best measure, derive, use, and test motif models. Further, there was no commonly accepted set of PWMs that could serve as a reliable baseline. This deficit complicates a fair assessment of alternative PWMs or comparison of PWMs to more complex models, able to account for correlations of nucleotides within binding sites, even for widely studied TFs with well-described DNA binding specificities. Particularly, in some applications, the complex models can fall behind carefully selected PWMs[49], but without the commonly accepted baseline, it remains problematic to provide reliable quantitative estimates of the models' added value.

In our study, we used multiple motif discovery tools and multiple performance metrics across multiple experimental platforms. We identified motifs that were ranked consistently high by all metrics and across multiple replicates and individual test datasets, such as SELEX cycles or test data from alternative PBM normalization strategies. This approach allowed us to avoid prioritizing motifs that received high scores in a single benchmark by chance: different performance metrics were positively correlated but agreed imperfectly and produced different motif rankings (Supplementary Fig. 12). Thus, we consider that the Codebook MEX motif set and the underlying data provide a valuable resource for further development of DNA motif discovery tools. Of note, in our study, we employed several advanced motif discovery methods, including gkmSVM, ProBound, and ExplaiNN. However, while testing them against classical tools, we reduced their efficacy by converting their results to simple PWMs. Therefore, it remains of interest to perform a dedicated study of advanced motif models using the MEX PWMs as a baseline.

A related problem arises with human curation of experimental datasets based on PWM-represented motifs: there might be TFs with intricate

DNA-binding specificity that cannot be captured by PWMs, making it impossible to properly assess and approve the dataset. Finally, by design, we did not balance the starting motif sets across tools, allowing multiple candidate motifs to enter the benchmarking pipeline. Although running a conventional tool multiple times and collecting multiple alternative motifs might be a common practical scenario when performing an exploratory analysis of TF-DNA binding specificity, this is not a fully suitable approach in terms of benchmarking. Yet, in our study, larger motif sets did not enable the respective tools to occupy the podium despite having more attempts (Figs. 1B–D and 2A, B, Supplementary Fig. 1).

Dimont was the most versatile tool for PWM motif discovery, achieving the best performance on the entire experimental dataset, while ChIPMunk and ProBound could, in some cases, compete with Dimont for ChIP-Seq/GHT-SELEX and HT-SELEX. Classical tools such as MEME and HOMER rarely gave the best-performing motifs but often provided stably good results, as the gap in absolute performance from the best motifs was neither borderline nor dramatic. The most notable difference of Dimont compared with most alternative motif discovery approaches is that the PWM model in Dimont is optimized for a discriminative objective function numerically instead of using count-based statistics. Depending on the experimental method, "discriminative" may refer to distinguishing bound from unbound sequences, or to reconstructing a continuous scale of signals related to binding strength. Hence, Dimont optimizes its PWM using an objective function (though on the training data), which is in some sense related to the performance metrics that have been used for evaluation on the test data in the benchmark. Despite shared basic preprocessing, classical tools designed in the age of smaller datasets are prohibitively inefficient with larger data. Thus, we supplied these tools with subsets of peaks or reads, while advanced tools such as Dimont were able to utilize the complete training data. Analyzing larger sequence sets has the potential to yield better motifs, but in this study, we intentionally mimicked the scenarios of typical practical usage. A related, expected, but important, observation is about STREME, which is computationally efficient, but did not stand up to MEME in terms of the quality of the resulting motifs, and this trade-off between speed and accuracy should be taken into account if the motifs are to be used in any downstream analysis such as TFBS prediction.

It is important to count failures as well as successes. While most of the tools performed well on average, particular combinations of TFs, tools, and types of experiments could be more problematic, and such cases are difficult, if at all possible, to detect ab initio. Importantly, our study benefited greatly from multiple types of experiments for a single TF, which is quite rare in practice, where researchers usually limit themselves to a single assay. In real-life scenarios, the success rate of motif discovery will be even lower and simultaneously harder to assess, as here we discarded more than half of the experiments as "non-approved", but human curation would have been much more error-prone without the availability of data from multiple experimental platforms. Further, our setup allowed detecting and filtering common artifact motifs, which could not be reliably pinpointed without systematic cross-platform data on multiple TFs. We consider the artifact filtering an important step of the pipeline, but do not expect the particular set of template motifs to be directly applicable to other studies, as the appearance of particular artifacts varied between platforms and some common patterns (like NFY- and ETS- motifs) in some scenarios can represent meaningful motifs.

Our large-scale motif discovery and benchmarking efforts also highlighted the widely debated topic of whether innate TF binding specificity or genomic context (e.g., DNA bases flanking the directly contacted binding site) contribute more to the genomic binding profile of a TF. In our observations, the motifs learned from synthetic sequences were highly transferable to genomic regions, thus reinforcing the idea that innate DNA binding specificity is localized, highly sequence dependent, and can be efficiently learned and modeled from diverse experimental data, including synthetic DNA sequences. The property of many TFs to recognize several motif subtypes can be addressed by "ensembling" PWMs in the random forest classifier, which was able to achieve better performance when trained and tested on GHT-

SELEX and ChIP-Seq peaks, suggesting that cell type-agnostic, reliable binding profiles can be generated in silico from DNA sequence alone[25]. However, it was significantly more difficult for models to generalize beyond the particular experimental platform, even though the underlying motifs were built from diverse experiments. Thus, we expect transferability and generalization to remain major challenges for machine learning applications in advanced TFBS modeling, and their success will likely depend on creative approaches for data integration from multiple experiment types.

Despite the richness of the underlying data and multiple benchmarking protocols involved, we caution against overestimating the potential of the PWM as a model. First of all, many TFs, especially those with multiple zinc fingers or several DNA-binding domains of different classes, can recognize alternative motifs or motif subtypes, which are impossible to capture with a single PWM model. Thus, with PWM-based motif discovery, some motif subtypes can be mixed or missing, and the benchmarking results might be biased towards either primary subtypes or mixtures of alternative patterns. Next, the scoring method is important: as in Ambrosini et al.[17], most of our benchmarking protocols used the sum-occupancy scoring[16], which effectively accounted for the contribution of multiple binding sites. The ChIP-Seq and GHT-SELEX benchmarks with the best log-odds PWM hits were less sensitive, resulting in smaller differences between tools (Supplementary Fig. 4), particularly, Dimont lost its edge. In this sense, selecting the best motif discovery tool is less crucial if the goal is to pinpoint the single best binding site within a known binding region.

Of note, only 236 of the 394 putative TF proteins examined in our study had at least one approved experiment. We explored multiple motif discovery tools but used a deliberately conservative human curation protocol. Thus, we encourage the community to further explore the remaining data with advanced models or more sophisticated preprocessing strategies, as it should be possible to successfully discover motifs from some of the non-approved experiments, as has been shown for SMiLE-Seq[26].

The most recent attempt to rigorously catalog human DNA-binding TFs[50] assigned the respective GO term to 1435 human proteins, requiring strict experimental evidence of both the role played by the particular protein in transcriptional regulation and its DNA-binding specificity. The set of motifs curated and rigorously analyzed in our study provides strong evidence for DNA-binding specificity for 54 additional proteins, which should now become prime candidates for genuine sequence-specific DNA-binding human TFs.

## Methods
### Workflow overview
The study relies on results of diverse experiments performed by Codebook to assess DNA binding specificity of human transcription factors: ChIP-Seq (CHS), HT-SELEX with random DNA (HTS), HT-SELEX with genomic DNA (genomic HT-SELEX, GHTS), protein-binding microarrays (PBMs), and SMiLE-Seq (SMS), see Jolma et al.[23]. For PBMs, two results from two alternative designs (ME and HK) were available. For HTS and GHTS, there were three distinct experimental designs, which differed in target protein production, namely, in vitro transcription (-IVT), GFP-tagged IVT (-GFPIVT), and cell lysate (-Lysate). For SMiLE-Seq, in addition to the new Codebook experiments, we included 27 previously published SMS datasets as additional positive controls[39]. In comparison to the main Codebook study[23], we additionally used new SMiLE-Seq data for TFE3 (positive control) and ZNF346 (SMiLE-Seq negative control, RNA-binding protein with non-specific DNA binding), explicitly considered the data for GFP (negative control), and excluded BAZ2A and REXO4 at the earlier stage as they did not yield sufficient sets of ChIP-Seq or GHT-SELEX peaks and their other experiments were also deemed unsuccessful. Some ChIP-Seq and GHT-SELEX experiments were deemed non-approved prior to benchmarking and curation as they did not yield a sufficient number of peaks: we required that the experiment must yield 50 or more technically reproducible peaks (see the ChIP-Seq peak calling details below) at the even-numbered autosomes used for motif benchmarking, see below. Also, we considered technical sequencing replicates independently as they yielded overlapping but not identical peak sets. The complete starting set contained results from

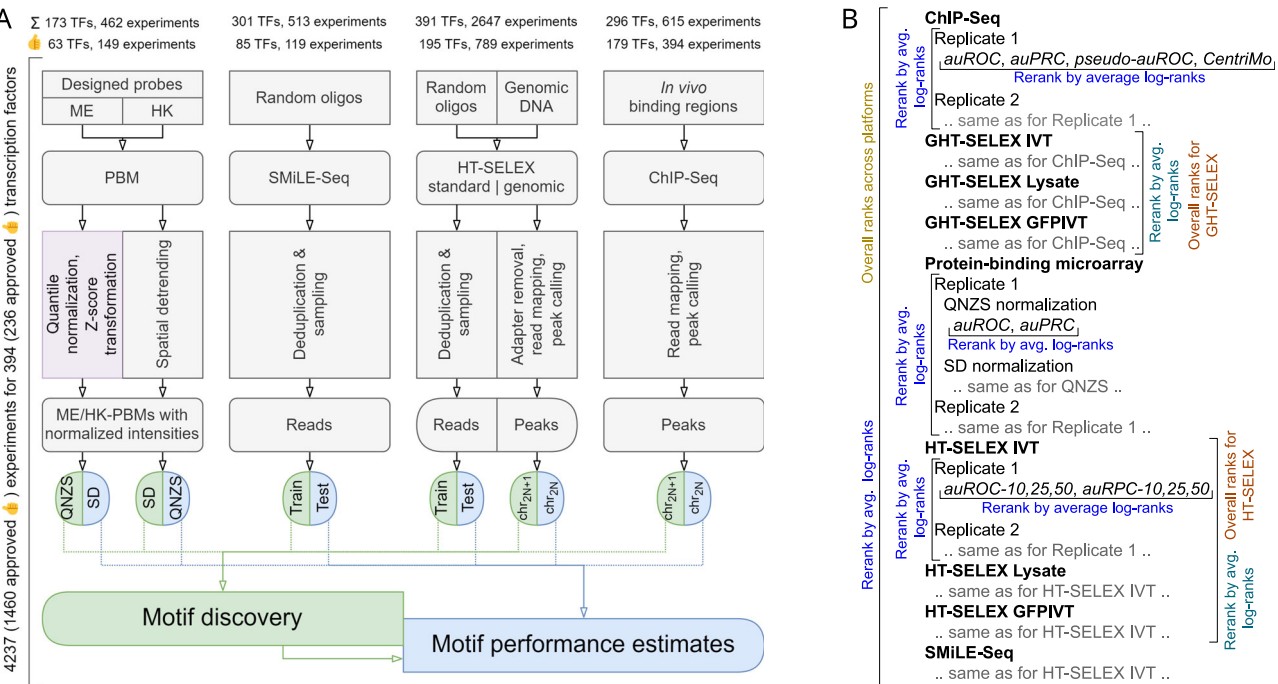

**Fig. 5 | Schematics of the underlying workflows. A** Experimental data preprocessing and generation of train-test data slices for motif discovery and benchmarking. For (G) HT-SELEX, multiple SELEX cycles are counted as a single experiment. **B** Scheme of the hierarchical rank aggregation.

4237 experiments (including technical sequencing replicates) for 394 proteins, see Supplementary Data 1.

The general workflow of the study is shown in Fig. 1A. Briefly, the experimental data were preprocessed, split into training and validation data, and passed to the first round of motif discovery with nine different software tools (see details below). While all tested tools were generating PWMs in the end, they represented two distinct categories. (1) Classic probabilistic and enumerative motif discovery tools: Autoseed[31], ChIPMunk[30], HOMER[29], MEME[28], and STREME[32]. (2) Advanced tools utilizing probabilistic discriminative learning (Dimont[33]), protein sequence information (RCade[35]), and modern machine learning techniques (ExplaiNN[34], gkmSVM[36] followed by GkmExplain[51]). For motif benchmarking, depending on the particular protocol (see below), we used sum-occupancy scoring with PFMs or best hits of log-odds PWMs.

Results of the 4237 experiments for 394 proteins were used in the first round of motif discovery with nine tools. The motifs were benchmarked and their logos, along with quantitative performance metrics for recognizing binding sites across multiple datasets, were used for expert curation of the datasets, see the details below. The data from 1460 curated and approved experiments for 236 TFs were then used for the second round of motif discovery with the addition of ProBound[39] and extra motifs generated by the conventional tools (MEME, HOMER, RCade, STREME) using alternative settings, see below. Motifs from the second round were also benchmarked and put together with the results from the first round for the curation-approved experiments. Of note, motifs highly similar to common experimental artifacts were filtered before benchmarking (see below).

The Codebook Motif Explorer (MEX) website (https://mex.autosome.org) provides motifs from both rounds of motif discovery and all datasets, while only motifs originating from the curation-approved experiments were included in the comparison of tools and experimental methods presented in this study. The motif logos were plotted with *drawlogo* (https://pypi.org/project/drawlogo/), which scales the nucleotide letters based on discrete information content[30] with inflated pseudocounts to achieve visual clarity for low-information content motifs. The software implementation of the data processing pipeline is available on GitHub (https://github.com/autosome-ru/greco-bit-data-processing).

## Experimental data preprocessing

An overview of the experimental data and preprocessing pipeline followed by motif discovery and benchmarking is shown in Fig. 5A. Preprocessing protocols for particular platforms are described in detail below.

**HT-SELEX and SMiLE-Seq.** No special preprocessing was performed for HT-SELEX or SMiLE-Seq data (FASTQ). However, in HT-SELEX and SMiLE-Seq, the binding sites may overlap the constant parts of the oligonucleotides that were physically present during the binding experiments, i.e., the binding sites may include parts of the primers and/or barcodes, which vary from experiment to experiment. Therefore, this information was saved in file names and metadata and then explicitly made available during motif discovery and benchmarking.

The sequence design of HT-SELEX data was the following:

5' ACACTCTTTCCCTACACGACGCTCTTCCGATCT [BAR1] (40 N) [BAR2] AGATCGGAAGAGCACACGTCTGAACTCCAGTCAC 3' where BAR1 and BAR2 were experiment-specific barcodes, and 40 N was a 40 bp random insert.

The sequence design of the Codebook SMiLE-Seq data was the following:

5' CGTCGGCAGCGTCAGATGTGTATAAGAGACAG [BAR1] (40N) CTGTCTCTTATACACATCTCCGAGCCCA 3' with a 40 bp random insert. The sequence design of previously published SMiLE-Seq data was the following: 5' ACACTCTTTCCCTACACGACGCTCTTCCGATCT [BC-half1] (30N) [BC-half2] GATCGGAAGAGCTCGTATGCCGTCT TCTGCTTG 3' with a 30 bp random insert.

**Protein-binding microarrays.** The Codebook data were obtained from two PBM designs, ME and HK[23], which were preprocessed separately to account for systematic biases, for example, from the arrangement of probes on the microarray. We used two preprocessing strategies:
(1) QNZS, quantile normalization followed by Z-scoring. Here, log-transformed probe intensities underwent quantile normalization to make the signal distributions of each array identical. Next, the intensity of each probe underwent a Z-score transformation, with the mean and std. dev. assessed for each probe separately based on its intensities across all available PBMs.

(2) SD, spatial detrending with window size $11 \times 11$ as tested in Weirauch et al.[19]. For motif discovery, we also performed quantile normalization after spatial detrending (SDQN) to have a uniform normalized scale for all datasets.

The software implementation of the procedures is available at https://github.com/autosome-ru/PBM_preprocessing.

**ChIP-Seq and genomic HT-SELEX.** The analysis of both ChIP-Seq and GHT-SELEX data was performed using the unified GTRD ChIP-Seq pipeline[52]. Both for ChIP-Seq and genomic HT-SELEX (GHT-SELEX, GHTS), the FASTQ read alignment was performed with *bowtie2* (2.2.3, default parameters and fixed `--seed 0`). For paired-end reads, we additionally specified `--no-mixed --no-discordant --maxins 1000`. Reported alignments were filtered by MAPQ score with *samtools* `-q 10`. For paired-end data, we additionally marked and removed PCR duplicates with *Picard MarkDuplicates*. Specifically, for the genomic HT-SELEX data, before read mapping, we performed adapter trimming with *cutadapt* (version 1.15, AGATCGGAAGAGC as the adapter sequence: `-a AGATCGGAAGAGC -A AGATCGGAAGAGC -o out.-R1.fastq.gz -p out.R2.fastq.gz in.R1.fastq.gz in.R2.fastq.gz`).

To achieve a balanced sequencing depth between experiments and controls and reduce computational load, peak calling was performed against randomly sampled control data (10% of the total pooled set of control reads from the matching batch, sampling performed after the alignment step). For ChIP-Seq, the input DNA samples were used as the control. For genomic HT-SELEX, the zero-cycle unselected reads were used as control. Paired-end control data was prioritized for paired-end ChIP-Seq when available in the same batch.

For peak calling, four tools (*macs2* 2.1.2, *pics* https://github.com/Biosoft-ru/cpics, *gem* 2.5, *sissrs* 1.4) were executed with default settings, except for *macs2*. For the latter, for single-end reads, the expected fragment length `$frag_len` was estimated with a strand cross-correlation approach (*run_spp.R* from the ENCODE pipeline dated Aug 29, 2016, https://github.com/kundajelab/phantompeakqualtools). Next, *macs2* was executed with `--nomodel --extsize $frag_len` for single-end read alignments. For paired-end reads, we ran *macs2* in the paired-end mode (`-f BAMPE --nomodel`). Single-end peak callers (*pics*, *gem*, *sissrs*) were executed on paired-end data using alignments of the first reads in a pair (*samtools* `-F 128 paired.bam`). The primary peak calls for each dataset were obtained with *macs2*. Next, technically reproducible *macs2* peaks were selected by ensuring a non-empty overlap with any of the peaks from other peak callers (*pics*, *sissrs*, *gem*). For GHT-SELEX, the peak calling was performed separately for reads originating from each cycle. The resulting files follow the *macs2* peak call format (narrowPeak) with an additional column listing supporting evidence from our peak callers. The resulting peaks were sorted by chromosome and coordinates.

## Train-test data splits and benchmarking protocols

In this study, we focused on a fair assessment of motif performance. Thus, for each experiment, we have generated separate non-overlapping training and test datasets. The only exception was PBM data, where we allowed a criss-cross training-test for different normalization strategies (SD and QNZS), i.e., for a particular PBM, QNZS data were applicable for testing SD-derived motifs and vice versa. The experiment-specific benchmarking train-test splits and protocols are described below. The implementation is available on GitHub (https://github.com/autosome-ru/motif_benchmarks). For all motif discovery tools, the resulting motif model was a matrix of positional nucleotide counts or a matrix of normalized nucleotide frequencies.

**Benchmarking with PBM probe intensities.** PBM data were used to assess motif performance using the binary classification of positives (specifically bound probes) and negatives (the rest of the probes): area under the receiver operating characteristic (auROC) and area under the

precision-recall curve (auPRC) were computed with Jstacs[53]. For SD-preprocessed PBMs, the probes passing mean $+ 4$ std. dev. intensity threshold were designated as positives as in the PBM-centric DREAM challenge of Weirauch et al.[19], see Online Methods, "AUROC of probe intensity predictions" section. For QNZS-normalized PBMs, probes with Z-scores above 4 were considered positives. In case these rules provided fewer than 50 positives, a minimum of 50 top-scoring probes was used instead. Motif scanning for this metric used the sum-occupancy scoring with PFMs. During scanning, the first 6 bps of the static linker sequence were concatenated to the unique sequence of each probe.

**Benchmarking with ChIP-Seq and GHT-SELEX peaks.** The train-test split for peaks data was performed using complete chromosome holdout: peaks at odd-numbered autosomes were designated for model training, and peaks at even-numbered autosomes for testing. Only experiments yielding 50 or more peaks in the test data were used for benchmarking, and only datasets with 50 or more peaks in the training data were used for motif discovery. Individual GHT-SELEX cycles were considered separately. Three different peak-based benchmarks were used.

(1) The Orenstein-Shamir[16] setup for binary classification of positives against neighboring negative regions, yielding area under the receiver operating characteristic (auROC). We used the approach implemented in Ambrosini et al.[17] with the following modifications: up to the top 1000 peaks were used to generate positives as 250 bp-long $[-124, +125]$ regions around the peak summits; for each positive peak, 250 bp regions located 300 bp upstream and downstream from the original peak summit were included in the negative set. Motif scanning for this metric used the sum-occupancy scoring with PFMs. The area under the precision-recall curve (auPRC) was estimated in addition to the auROC.

(2) The asymptotic pseudo-auROC as in HOCOMOCO v11[54]. This method compared the top-scoring PWM motif hits in positives against the asymptotic estimate for random sequences of the same lengths and dinucleotide composition as in the positive sequence set. Positive regions' lengths were standardized by taking ±150 bp around the peak summits, and up to 1000 first peaks from the test data (reproducible peaks sorted by chromosome and coordinate, see above) were used. Motif scanning for this metric used best-hit log-odds scoring, and PFMs were transformed to log-odds PWMs in the following way[55]:

```
count_ij = count × freq_ij
pseudocount = log(max(2, count))
pwm_ij = log(
(count_ij + 0.25 × pseudocount) / (0.25 × (count +
pseudocount)))
```

where `count_ij` is the *i,j*-th element of the matrix of non-normalized nucleotide counts. For ChIPMunk, `count` was set to the actual number of aligned words. For other methods yielding normalized PFMs, `count` was set to 100.

(3) CentriMo[38] motif centrality measure ($-\log$-$E$-value) for motif hit locations against peak summits. For motif scanning, CentriMo performs the PFM-to-PWM transformation internally. In the case the run with default parameters technically failed to provide output (e.g., due to a few sufficiently high-scoring sites), we reran CentriMo with `--score 1 --use-pvalues` to allow it to consider low-scoring motif occurrences.

**Benchmarking with HT-SELEX and SMiLE-Seq reads.** For each cycle, reads were separated into the train and test datasets in a 2 to 1 ratio. At the benchmarking stage, the reads from different cycles (for HT-SELEX) were pooled together. We randomly sampled a maximum of 500,000 unique reads per dataset and used the Orenstein-Shamir benchmarking protocol (sum-occupancy scoring with PFMs) as in Ambrosini et al.[17] with 10%, 25%, or 50% of top-scoring reads to be designated as positives

for each tested PWM. In addition to auROC (as in the original protocol), we also computed auPRC.

## Identifying best-performing motifs

For each motif and each test dataset, we computed several performance measures. To identify the best-performing motif, we performed hierarchical rank aggregation as suggested in the DREAM-ENCODE challenge (https://www.synapse.org/#!Synapse:syn6131484/). We ordered the motifs by achieved performance for all combinations of benchmarks and performance metrics and calculated the geometric mean of the ranks, followed by re-ranking (Fig. 5B):

first, across different metrics of a single benchmark (e.g., auROC and auPRC), then across variants of benchmarking settings (e.g., HT-SELEX benchmark with 10%, 25%, or 50% top hits taken as positives), and then across different benchmarks (e.g., for ChIP-Seq, Orenstein-Shamir classification performance and CentriMo motif centrality);

next, across independent experiments of the same type (experimental replicates) and technical sequencing replicates, which were available for select ChIP-Seq datasets.

This provided the best motif for a TF for a particular type of experimental data. Next, aggregation across all data types was performed to identify the best motif for each TF in terms of the overall performance across experiment types.

The procedure was performed twice: once for the complete data (precuration) and once for the curation-approved experiments and respective motifs. The results of both variants are available online in Codebook MEX, and the data from curation-approved experiments were used for detailed analysis of motif performance in this study. The software implementation of the ranking procedure is available on GitHub (https://github.com/autosome-ru/greco-bit-data-processing).

Note, that in the overall ranking (Fig. 1), individual HT- and GHT-SELEX types (Lysate, IVT, GFPIVT) were considered as independent platforms, while in the subsequent analysis (Figs. 2 and 3) they were considered together.

**Harmonizing benchmarking measures.** The range of performance measures depends not only on the benchmarking protocol but also on the TF and the particular experimental dataset. To make the benchmarking metrics comparable and to obtain a common scale in Fig. 2, we applied a linear transformation to project the raw values of the benchmarking metrics (auROC, auPRC, etc) into the [0;1] range for each <TF, dataset, benchmark, metric> combination independently, where 0 corresponded to the lowest achieved value of the worst-performing motif, and 1 was the highest achieved value of the best-performing motif. At the cost of direct interpretability of metric score differences, this allowed quantitative analysis of metrics across TFs and experiment types.

## Dataset curation procedure

The availability of multiple types of experiments and replicates facilitated the comprehensive manual curation of datasets in terms of the consistency of DNA specificity patterns discovered from different types of experiments. In addition to the general consistency of motifs (e.g., visual similarity of logos) derived from different types of experiments, a major argument for dataset approval was the cross-experiment motif performance, i.e., motifs trained on one and high-scoring on the other type of experimental data were supporting approval of both the source and the benchmarking dataset. To simplify curation, we have annotated the motifs with the closest known patterns from HOCOMOCO v11[37] using MACRO-APE[55], this annotation is available in MEX from the curation stage.

During curation, each experiment was examined by at least one junior and two of three senior curators (A.J., I.V.K., and T.R.H.). The key features to analyze were the best-performing motifs, their datasets of origin, types of experimental data, and absolute achieved performance metrics. Cases with discordant approved/non-approved curation labels

were individually rechecked, discussed, and resolved by two senior curators (A.J. and T.R.H.).

Of note, traditional quality controls, such as ENCODE-style metrics for ChIP-Seq data, were also taken into account during the curation and are available in the experiment metadata in MEX. However, there were cases when formally poor QC experiments yielded the proper motifs or ranked high proper motifs from other datasets, allowing approval of such datasets.

## Filtering artifact signals

The motif benchmarking and dataset curation could be complicated due to common artifacts related to particular types of experiments, where highly similar motifs were observed in a large number of experiments for different proteins. Some of these artifacts, such as the ACGACG sequence observed in HT-SELEX, match constant flanking regions of the SELEX ligands and are likely derived from the enrichment of partially single-stranded ligands, whereas others are only seen in the lysate-based experiments and correspond to endogenous TFs (such as NFI and YY1) that are highly expressed in HEK293 cells.

To reduce the overall impact of these widespread artifact signals, we manually assembled a catalog of artifact motifs during the curation stage (downloadable from the MEX website and from the MEX Zenodo repository[40–42]). Next, we filtered the whole motif collection by comparing the motifs against the catalog using MACRO-APE[55] (with the motif $P$-value threshold $5 \times 10^{-4}$ and -d 10). Motifs with Jaccard similarity ≥0.15 between high-scoring word sets[55] were filtered out. We also filtered the motifs scoring below $P$-value $10^{-4}$ in constant flanks of HT-SELEX, genomic HT-SELEX (except the generally less noisy GFP-tagged IVT design), or SMiLE-Seq in a primer/barcode-dependent way. Of note, ETS-related motifs were not filtered for ETS-related "positive control" TFs (ELF3, FLI1, and GABPA).

## Motif discovery tools and data tool-specific data processing

The first round of motif discovery was focused on applying a diverse set of tools to the complete Codebook data, and its results were used for the dataset curation, as described above, to include only the data from approved experiments in the downstream analysis of motif performance. The second round of motif discovery was to generate more motifs from the curation-approved experiments by employing popular motif discovery tools used in the first round with alternative settings and on more data types, while including motifs yielded by the recently published ProBound software[39]. Of note, not all motif discovery tools were applicable in a ready-to-use fashion to all data types, and thus, extra preprocessing was necessary to use the tools on data types for which they were not designed in the first place.

## ChIPMunk

ChIPMunk is a PWM-based greedy optimization algorithm suitable for processing thousands to tens of thousands of sequences with or without positional prior such as ChIP-Seq peak summit location.

**Motif discovery from ChIP-Seq and GHT-SELEX peaks. Data preparation.** ChIP-Seq and GHT-SELEX peak calls were sorted by peak height and the top 500 and 1000 regions, respectively, were taken for subsequent analysis. 301-bp-long regions centered on the peak summits were extracted for motif discovery in ChIPMunk "peak" mode by specifying 150 as the relative peak summit location.

**Motif discovery from peaks.** The ChIPMunk launcher script was executed with the following parameters:

```
ruby   run_chiphorde8.rb   <motif_name><start_
motif_length><stop_motif_length> filter yes 1.0 m:
<input_filename> 400 40 1 2 random auto <shape>.
```

The motif discovery was performed three times using (1) 21 to 7 motif lengths range, no motif shape prior; (2) 15 to 7 lengths range, no motif shape prior; (3) 7 to 15 lengths range, "single-box" motif shape prior. For Zinc-finger TFs, we expected longer motifs and used alternative settings: (1) 25 to 7 lengths range, no motif shape prior; (2) 7 to 21 lengths range, "single-box"

motif shape prior. Using the "filter" strategy, ChIPMunk performs filtering of the initially found primary motif hits and could yield a secondary alternative motif.

**Motif discovery from HT-SELEX, GHT-SELEX, and SMiLE-Seq reads. Data preparation.** For (G)HT-SELEX, we pooled reads across all cycles of an experiment or for the terminal 3+ cycles only; the complete training data was used for SMiLE-Seq. To account for binding sites overlapping constant parts (technical segments) of the oligonucleotides, the reads were extended as $5'-NNX_1<read>X_2NN-3'$, where $X_1$ and $X_2$ belonged to the technical segments and thus were the same across the sequences from a particular experiment. Singleton sequences found only once in the pooled dataset were excluded. Next, 5-mer enrichment against the dinucleotide shuffled control was computed with the custom script (https://github.com/autosome-ru/HT-SELEX-kmer-filtering). For each dataset, we gathered 500, 1000, and 2500 top-enriched sequences for motif discovery with dinucleotide ChIPMunk[56] and 10,000 sequences for ChIPMunk; all available sequences were used in the case of fewer sequences than that number available. For dinucleotide ChIPMunk, the standard position weight matrices were constructed from the resulting multiple sequence alignments.

**Motif discovery from reads.** The ChIPMunk was used via Ruby launcher with the following parameters specifying 7 to 25 motif length range:

```
ruby <motif name> 7 25 yes 1.0 s:<sequence set file> 100
10 1 2 random auto flat yes
```

**Motif discovery from PBM data. Data preparation.** For each microarray, we used the sequences of the top 1000 probes ranked by normalized signal intensity, skipping the flagged probes[57]. The sequences were taken without the linker flank.

**Motif discovery from PBM probes.** The dinucleotide version of ChIPMunk was used with the following parameters:

```
java -cp chipmunk.jar ru.autosome.di.ChIPMunk 6 16 y
1.0 s:<sequence set file> 400 40 1 2 random auto flat
```

The single-nucleotide position count matrices were constructed from the multiple sequence alignments.

## Dimont

Dimont is a motif discovery algorithm that allows for modeling binding motifs using Markov models in general and PWMs in particular. Dimont has been designed for using all available sequences from binding experiments (e.g., all ChIP-Seq peaks), where each sequence is associated with a measure of confidence that this specific sequence is bound (e.g., peak statistics from ChIP-Seq experiments), which are converted to soft class labels (bound vs unbound) with assay-specific formulas. The objective function of Dimont (maximum supervised posterior) optimizes the concordance of these soft labels and motif-based scores using gradient-based numerical optimization, i.e., Dimont tries to find the motifs that explain the soft labels best. Dimont-HTS is a variant of the Dimont algorithm with an HTS-specific weighting schema for the soft labels and an adapted initialization strategy.

Motif discovery using Dimont requires an input set of sequences, which are complemented by a sequence-specific "signal" annotation, which indicates the confidence that a specific sequence is bound by the TF of interest. Signal values are converted to soft labels internally using a rank-based method[33], where a value of 1 indicates perfect confidence that a sequence is bound and 0 indicates perfect confidence that a sequence is not bound by the TF. Dimont then aims at finding the motif that explains the soft labels best, i.e., that yields high scores for sequences with soft labels close to 1 and low(er) scores for sequences with soft labels close to 0. This is achieved by maximizing the supervised posterior of the soft-labeled input data[33]. In the following, we describe for the specific data types how sequences were extracted, how "signal" values were defined, and how these were used for motif discovery using Dimont.

**Data preparation for ChIP-Seq and GHT-SELEX peaks.** All ChIP-Seq and GHT-SELEX peaks in the training set were considered, and 1000-bp-long regions around the peak centers were extracted together with the corresponding peak statistics (column 7 of the peak list) and stored in FastA format. The peak statistic was used as a "signal" annotation in the FastA headers of the extracted sequences and is subsequently used for determining weights in the Dimont learning procedure.

**Data preparation for PBM probes.** For each probe, the unique probe sequence and the first 6 bp of the static linker sequence were concatenated and extracted together with the mean signal intensity value of the corresponding probe and stored in FastA format. The mean signal intensity was used as a "signal" annotation in the FastA headers of the concatenated sequences.

**Data preparation for HT-SELEX reads.** First, reads from all HT-SELEX cycles were extracted and stored in FastA format using the HT-SELEX cycle as a "signal" annotation. Then, reads across all cycles of an HT-SELEX experiment for a specific TF were sub-sampled to at most 400,000 reads, while sampling reads from the different cycles such that the distribution across cycles was as even as possible.

**Data preparation for SMiLE-Seq reads.** Reads from each SMiLE-Seq experiment were extracted and stored in FastA format. For a specific SMiLE-Seq experiment, all reads were assigned a "signal" annotation of 1. These were complemented with a sub-sample of one-fifth of the original reads from all other SMiLE-Seq experiments from the same batch (barcode) but for other target TFs, which were assigned a "signal" annotation of 0.

**Motif discovery from ChIP-seq and GHT-SELEX peaks, PBM probes, and SMiLE-Seq reads.** For sequences generated from ChIP-seq and GHT-SELEX peaks, PBM probes, and SMiLE-Seq reads, Dimont was executed with default parameters with a few minor exceptions. For ChIP-seq and GHT-SELEX, the initial motif width was set to 20 (imw = 20). For PBM probes and SMiLE-Seq, the initial motif width was set to 10, and masking of previous motif occurrences was switched off (imw=10 d=false), while the weighting factor was set to 0.05 (w = 0.05) for PBM probes and 0.5 (w = 0.5) for SMiLE-Seq. The first two motifs reported by Dimont were used for further analyses.

**Motif discovery from HT-SELEX reads.** For HT-SELEX, two alternative strategies were used. In the first approach, the cycles stored as "signal" values were converted to soft labels based on an enrichment factor E as $E^{cycle-max(cycle)}$. Aside from the definition of soft labels, Dimont was started with default parameters. The second approach (Dimont-HTS) was more specifically tailored to HT-SELEX data. Here, the motif initialization step of Dimont was based on 10-mers identified by a re-implementation of the Z-score proposed by Ge et al.[58] and filtered for redundancy using a minimum Huddinge distance[31] of 2. The determination of soft labels was based on a cycle-specific and a sequence-specific weight, interpolating linearly between adjacent cycle-specific weights. The cycle-specific weight was determined from the relative number of unspecific sequences in each HT-SELEX cycle using a re-implementation of the method proposed by Jolma et al.[15]. Within each cycle, sequence-specific weights were determined based on the ranks of 8-mer occurrences among all sequences of a cycle. Sequence-specific weights were defined as the maximum relative rank of an 8-mer occurring in a

sequence divided by the maximum rank across all sequences. Besides the adapted initialization strategy and determination of soft labels, the initial motif width was set to 20 (imw = 20). Again, the first two motifs reported by Dimont were used for further analyses. The respective code is available at GitHub (https://github.com/Jstacs/Jstacs/tree/master/projects/dimont/hts).

## ExplaiNN

ExplaiNN is a fully interpretable and transparent sequence-based deep learning model for genomic tasks that combines the powerful pattern recognition capabilities of convolutional neural networks with the simplicity of linear models.

**Data preparation**. To construct the training and validation datasets, each experiment was processed separately. Additionally, for the PBM data, we avoided mixing data from different normalization methods. As a data augmentation strategy, we doubled the size of each training and validation set by including the reverse complement of each sequence.

**ChIP-Seq data**. The peaks were resized to 201 bp by extending each peak summit by 100 bp in both directions. Then, they were randomly split into training (80%) and validation (20%) sets using the train_test_s-plit function from scikit-learn (version 0.24.2, random splits were always performed in this manner)[59]. To avoid the need for negative samples during training, we retained the peak heights associated with each peak, thereby converting the training process into a regression task.

**HT-SELEX and GHT-SELEX data**. We treated cycles as independent classes, following the approach used by Asif and Orenstein[60], thereby avoiding the need for negative samples during training. The reads were then randomly split into training (80%) and validation (20%) sets while maintaining the proportions between reads from each cycle.

**PBM data**. The probes, including both the de Bruijn and linker sequences, were randomly split into training (80%) and validation (20%) sets. Since the training task involved regression (see below), negative samples were not required.

**SMiLE-Seq data**. A set of negative samples was obtained by dinucleotide shuffling using BiasAway[61] (version 3.3.0). Then, the original reads (positives) and the negative samples were combined and randomly split into training (80%) and validation (20%) sets, ensuring an equal proportion between positives and negatives.

**Model training**. All models featured the same architecture: 100 units and a filter size of 26. They were trained using the Adam optimizer[62] for a maximum of 100 epochs. An early stopping criterion was set to halt training if the validation loss did not improve after 10 epochs. We applied one-hot encoding to the input sequences, converting nucleotides into 4-element vectors (A, C, G, and T). The learning rate was set to 0.003, and we used a batch size of 100. During training, we employed three different loss functions, tailored to each data type.

**ChIP-Seq data**. ExplaiNN was configured to model the peak heights using the negative log-likelihood loss with a Poisson distribution of the target (PoissonNLLLoss class from PyTorch[63]).

**HT-SELEX, GHT-SELEX, and SMiLE-Seq data**. The modeling tasks for these data involved either multi-label classification (for SELEX) or binary classification (for SMiLE-Seq). As a result, we chose BCEWithLo-gitsLoss as the loss function (binary cross-entropy with sigmoid).

**PBM data**. ExplaiNN was applied to model normalized intensity signals, making the mean squared error (MSELoss) the appropriate choice for the loss function.

**Motif discovery**. Following the specifications from the ExplaiNN manuscript, for each model, we constructed a position frequency matrix (PFM) for each filter by aligning all 26-mers (26 bp-long DNA sequences) activating that filter's unit by ≥50% of its maximum activation value in correctly predicted sequences. Then, we transformed the resulting PFMs into position weight matrices (PWMs), setting the background uniform nucleotide frequency to 0.25, and clustered them based on their Tomtom similarity[64] using scripts from (https://github.com/vierstralab/motif-clustering). Finally, for each experiment (i.e., for each model), we returned the top 5 non-redundant PWMs (belonging to different clusters) based on their performance on the corresponding validation set.

## GkmSVM with GkmExplain

**Data preparation**. ChIP-Seq data were sorted based on the q-value, and the top 5000 peaks were taken. The top 5000 peaks were split into training and testing based on chromosomes (chr1 and chr3 used for testing). The peaks were extended by 100 bps on each side of the summit. For the negative set, for training, we used the fasta-dinucleotide-shuffle-py3.in from MEME suite[28] to generate dinucleotide shuffled peaks from our positive set data.

**Model training**. To train the gkmSVM model[51] we used the gkmtrain function from the LS-GKM package using the default parameters. LS-GKM is a version of gkmSVM[36], an SVM-based algorithm that utilizes gapped k-mers as features. LS-GKM is specifically optimized for processing and training on a large number of sequences efficiently. In the default parameters, the word length (-l) is 11, the gap (-d) is 3, and the kernel used is the center-weighted (wgkm) kernel.

**Motif discovery**. To generate motifs, we first generated importance scores and hypothetical importance scores using GkmExplain[36]. GkmExplain is a feature attribution technique applied to trained gkmSVM models that use a modified version of the integrated gradients method to determine the importance of individual nucleotides for the output label. GkmExplain has been shown to outperform[36] other feature attribution methods, such as deltaSVM[65] and in silico mutagenesis (ISM)[66]. Importance scores were generated from the test sequences and the train gkmSVM model using the command gkmexplain from LS-GKM package. The hypothetical importance scores were generated using the same command but with the parameter -m 1. To generate motifs from these importance scores, we ran TF-MoDISco (https://github.com/kundajelab/tfmodisco) with the following parameters (target_seq-let_fdr=0.2, sliding_window_size=21, flank_-size=10, min_passing_windows_frac=0.0005). TF-MoDISco uses importance scores derived from feature attribution methods to identify regions of high importance across sequences and clusters these recurring regions to generate motifs. Therefore, gkmEx-plain coupled with TF-MoDISco can be used to generate motifs from k-mer-based SVM models trained on our assays.

## HOMER

Homer is a motif discovery algorithm that uses word enumeration followed by the hypergeometric or binomial test to detect oligo enrichment in the input sequence[29]. HOMER then transforms the sets of detected oligos into PWMs via an iterative refinement and optimization process.

**Data preparation**. ChIP-Seq, HT-SELEX, GHT-SELEX, and SMiLE-Seq data were processed in the same way as for ChIPMunk (see above).

**Motif discovery**. We called the findmotifs.pl function with default parameters to find motifs using HOMER for all experimental assays. For the negative set required by HOMER we generated dinucleotide shuffled sequences using the fasta-dinucleotide-shuffle-py3.in script from MEME suite[28]. We also ran findmotifs.pl to find longer

motifs up to 30 bp by changing the -len parameter. The top 5 motifs outputted by HOMER for each set of parameters were used for analysis.

## MEME

The Multiple EM for Motif Elicitation (MEME) employs the expectation maximization (EM) technique to derive PWMs. The algorithm begins by detecting an initial seed motif, which is then iteratively optimized through EM steps, which continue until the PWM values stabilize or a predefined iteration limit is reached. MEME primarily operates using the Zero or One Occurrence Per Sequence model to discover ungapped motifs of fixed lengths.

**Data preparation.** Data for Chip-Seq, HT-SELEX, GHT-SELEX, and SMiLE-Seq were processed in the same way as for ChIPMunk (see above).

**Motif discovery.** We ran MEME-ChIP[67] on ChIP-Seq data and MEME[28] on data from other assays. Both MEME-ChIP and MEME were first run using default parameters. We additionally ran both MEME-ChIP and MEME using --maxw 30 and --minw 3 to account for the longer motifs of C2H2 Zinc-Finger TFs. The top 3 motifs outputted by MEME for each set of parameters were used in the downstream analysis.

## RCade

The Recognition Code-Assisted Discovery of Regulatory Elements (RCADE) algorithm is specifically built to uncover the binding preferences of the largest family of human transcription factors, the C2H2 zinc-finger proteins. By utilizing predictions from the DNA recognition code specific to Zinc Fingers[68], RCADE effectively infers the predicted binding motifs that are enriched in peaks compared to shuffled sequences.

**Data preparation.** Data for Chip-Seq, HT-SELEX, GHT-SELEX, and SMiLE-Seq were processed in the same way as for ChIPMunk (see above).

**Motif discovery.** We used RCADE2[35] (https://github.com/csglab/RCADE2) using default parameters to identify motifs for C2H2-Zinc Finger TFs across all the assays. The amino acid sequences of the entire TF used as a parameter by RCADE2 were downloaded from UniProt. The top motif outputted by RCADE2 was used in the subsequent analysis.

## STREME

STREME operates using a generalized suffix tree, a data structure similar to those used by tools like HOMER. STREME utilizes suffix trees to efficiently store input sequences and count matches between candidate PWMs (instead of oligos like Homer) and these sequences. After identifying potential motifs, STREME evaluates their enrichment in the input sequences using a one-sided Fisher's exact test against control sequences. Like MEME, STREME operates under the assumption of a Zero or One Occurrence Per Sequence (ZOOPS) model.

**Data preparation.** Data for Chip-Seq, HT-SELEX, GHT-SELEX, and SMiLE-Seq was processed in the same way as for ChipMunk (see above).

**Motif discovery.** We ran STREME[32] on data from all assays with two different sets of parameters. For the first run, STREME was run with default parameters. STREME was additionally used with --maxw 30 and --minw 3 parameters to account for the longer motifs of C2H2 Zinc-Finger TFs. The top 3 motifs outputted by STREME for each set of parameters were used for analysis.

## ProBound

**Data preparation.** For motif discovery using ProBound, k-mer count tables for each experiment were generated using all sequencing reads. The k-mer length was set to the entire probe length for SMiLE-Seq and HT-SELEX experiments. For GHT-SELEX, the probes were centered before extracting 40 bp of sequence (20 bp up and downstream of the center). Reads shorter than 40 bp were discarded. In the case of multi-round

SELEX experiments, columns indicating the round of enrichment were added to each count table. Since ProBound requires a sample of probe counts for an unselected input library, models could only be fit to SMiLE-Seq, GHT-SELEX, and HT-SELEX experiments. For SMiLE-Seq, input data were readily available for each experiment. For GHT-SELEX, input libraries were not matched to respective samples. Therefore, all input libraries were pooled, and 10,000 unique reads were sampled at random to build a global input count table. For HT-SELEX, a deeply sequenced input library was unavailable. To approximate the input library, the reads from "failed" experiments (those that showed no reliable probe enrichment after incubation with TFs) were combined, and 10,000 unique reads were sampled to create an approximate input count table. Note that this approach is not recommended in the original publication and may bias motif inference, e.g., the approximate input is nonetheless subject to non-specific binding preferences.

**Motif discovery.** ProBound was used with the following default optimizer settings: L2 regularizer weight of 0.000001 (lambdaL2 parameter), Dirichlet regularizer weight of 20 (pseudocount parameter), smallest improvement in likelihood required for a model variation to be accepted of 0.0002 (likelihoodThreshold parameter). All values were taken from the ProBound documentation of single-experiment transcription factor binding models. Other optimizer settings were left at default values, and no custom optimization was performed. Each experiment was analyzed with a single position-specific affinity matrix, which represents the change in binding affinity ($K_d$) for all point mutations with respect to the optimal reference sequence[69] binding mode with an initial size of 12 base pairs. For each experiment, a pair of models was developed—one incorporating the non-specific binding mode and the other one excluding it. In order to comply with the benchmarking pipeline, the energy logos produced by ProBound were first converted to position-specific affinity matrices (PSAMs) and then scaled to represent PFMs.

## Autoseed

Autoseed generates motifs with two sequence sets, e.g., for HT-SELEX, it uses a "signal" cycle (e.g., cycle 3 of an experiment) and a "background" cycle (e.g., cycle 2), setting an IUPAC base sequence (e.g., ACCGGAAGRN) as a seed and then obtaining a motif based on this sequence and all sequences that are within a parameter-specified edit distance from the seed (1, 2, or 3 edits). As in the previous work Nitta et al.[31], Autoseed was used to find Huddinge Distance-based local maxima for gapped 8-mers for combinations of a background and a signal cycle and to generate logos for these and heatmaps that display all possible spacing variants. Final motifs were generated manually by examining the Autoseed outputs to select optimal input parameters for motif generation.

## Random forest of PWMs

**Generating positive and negative datasets.** We specially adapted the MEX data to allow for unbiased training and testing of advanced models suitable for genomic TFBS prediction. To enable testing the transferability of predictions between data types, we considered TFs with at least one approved ChIP-Seq and at least one approved GHT-SELEX experiment. Separately for ChIP-Seq and GHT-SELEX, for each TF, the available peak sets were merged, and the peak summit locations of the overlapping peaks were averaged. The choice of train-test chromosome hold-out was the same as in the primary MEX benchmarking.

The "positive" sets (the bound regions) were created by extracting 301-bp-long regions centered at the resulting peak summits. Three alternative negative sets were generated:

(1) *Random* genomic regions (1:100 positive-to-negative class balance). Random regions were sampled from the genome matching the GC content distribution of the positive dataset.

(2) *Alien* peaks of other TFs (1:100 balance) were also sampled and extracted in the same manner to match the GC content distribution of the positive set. In the case of TFs with larger positive sets, all available

peaks were taken without GC matching if the 1:100 ratio was technically unachievable.

(3) *Shades* of true positive peaks, the neighboring upstream and downstream regions (up to 1:2 balance). For each positive peak, the summit of a fake upstream peak was uniformly selected from [−750 bp, −450 bp] interval relative to the true positive summit, and the summit of a fake downstream peak was uniformly selected from [450 bp, 750 bp] interval. In the end, the achieved balance was often closer to 1:1 as the regions to sample the shades were overlapping blacklisted regions (see below) or peaks of the same transcription factor.

For all types of negatives, we explicitly excluded positive regions (whole peaks), ENCODE blacklist regions[70], and any genomic regions with N nucleotides. The final Archipelago set of TFs included the data on 137 of 142 eligible proteins: TIGD4, TTF1, and ZBED5 models were not trained due to the presence of initially mislabeled datasets; CAMTA2 and FLYWCH1 models were trained but excluded from the in-depth analysis due to the underlying datasets having fewer "shades" than "positives".

### The Archipelago model
ArChIPelago, the arrangement of multiple position weight matrices with ChIP-Seq and machine learning for prediction of transcription factor binding sites, is a random forest on top of multiple PWMs. To construct Archipelago, separately for GHT-SELEX and ChIP-Seq data for each TF, we used the top 20 MEX PWMs best-performing at each replicate of ChIP-Seq and GHT-SELEX. ChIP-Seq-derived PWMs were not considered when training the model on GHT-SELEX and vice versa to prevent information leakage when evaluating model transferability.

The PWM predictions, i.e., the features for building the random forest, were obtained with SPRY-SARUS[71]: the log-odds PWM best hits in each sequence were identified using `--skipn --show-non-matching --output-scoring-mode score besthit`. The resulting feature matrix with class labels (1,0) was scale-transformed with `sklearn.-preprocessing.StandardScaler` from scikit-learn 1.3.2, and used to train a random forest classifier model with the following hyper-parameters: `['max_depth': 6, 'max_samples': 0.8, 'n_estimators': 100]`. The *random* negative set was used for model training. To estimate the Archipelago performance, we computed auROC and auPRC with PRROC R package[72] with three alternative negative datasets to reliably measure the model prediction quality. To select the top 5 motifs for logo visualization in Supplementary Fig. 9, we ranked all available motifs by the random forest feature importance estimated with `feature_-importances` of scikit-learn.

### Statistics and reproducibility
A significant fraction of the Codebook data was replicated (e.g., alternative constructs, alternative PBM designs, or alternative sequencing batches, see Supplementary Data 1 for details). For ChIP-Seq and GHT-SELEX, we used technically reproducible peaks only, as described above, but considered technical sequencing replicates independently as they yielded overlapping but not identical peak sets. One-tailed paired Wilcoxon test was used to assess the significance of performance differences between motifs derived from genomic and synthetic sequences in Fig. 3.

### Reporting summary
Further information on research design is available in the Nature Portfolio Reporting Summary linked to this article.

### Data availability
The interactive Codebook/GRECO-BIT Motif Explorer website is available online at https://mex.autosome.org. The complete set of MEX motifs and the benchmarking-ready Codebook data are available on ZENODO[40–42]. The Archipelago preprocessed data and trained models are available on ZENODO[73]. The numerical source data used for plotting the Figures are available at GitHub: https://github.com/autosome-ru/TSV_data_for_

Codebook_MEX_figures and https://github.com/autosome-ru/MEX-ArChIPelago.

### Code availability
The benchmarking protocols of Ambrosini et al.[17] are available on GitHub (https://github.com/autosome-ru/motif_benchmarks). The implementation of the data processing pipeline is available on GitHub (https://github.com/autosome-ru/greco-bit-data-processing). The software tools used in the study are listed in Supplementary Data 3. The code for Archipelago training and testing is available on GitHub (https://github.com/autosome-ru/MEX-ArChIPelago).

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

## Acknowledgements
We wholeheartedly thank the IT Group of the Institute of Computer Science at Martin Luther University Halle-Wittenberg for computational resources and, personally, Maximilian Biermann, for valuable technical support. We thank Gherman Novakovsky for providing useful feedback, Debashish Ray for assistance with database depositions, and Irina Eliseeva and Valery Vyaltsev for their help with the preparation of the graphical abstract. We thank members of the GRECO consortium and, personally, Martin Kuiper, for the encouragement and support of this project at its early stage in the form of dedicated workshops of the GREEKC COST action. This work was supported by the following: Canadian Institutes of Health Research (CIHR) grants FDN-148403, PJT-186136, PJT-191768, and PJT-191802, and NIH grant R21HG012258 to T.R.H.; CIHR grant PJT-191802 to T.R.H. and H.S.N.; Natural Sciences and Engineering Research Council of Canada (NSERC) grant RGPIN-2018-05962 to H.S.N.; Swiss National Science Foundation grant (no. 310030_197082) to B.D.; Marie Skłodowska-Curie (no. 895426) and EMBO long-term (1139-2019) fellowships to J.F.K.; DFG grant 514901783 (SFB 1664) to I.G.; NIH grants R01HG013328 and U24HG013078 to M.T.W., T.R.H., and Q.M.; NIH grants R01AR073228, P30AR070549, and R01AI173314 to M.T.W.; NIH grant P30CA008748 partially supported Q.M.; Canada Research Chairs funded by CIHR to T.R.H. and H.S.N.; Ontario Graduate Scholarships to K.U.L. and I.Y.; A.J. was supported by Vetenskapsrådet (Swedish Research Council) Postdoctoral Fellowship (2016-00158); The Billes Chair of Medical Research at the University of Toronto to T.R.H.; EPFL Center for Imaging; Institutional funding from EPFL; Resource allocations from the Digital Research Alliance of Canada; GTRD pipeline adaptation was supported by Russian Science Foundation grant 24-14-20031 to F.A.K.; A.Z. was supported by a personal fellowship from the Non-commercial Foundation for Support of Science and Education 'INTELLECT'; The allele-specific analysis was supported by the Ministry of Science and Higher Education of Russian Federation, grant number № 075-15-2025-014 (previously № 075-15-2024-666); Archipelago development was supported by assignment FFRW-2025-010.

## Author contributions
Writing—original draft, writing—review & editing: I.E.V., I.K., A.J., G.A., A.J.G, P.K., J.F.K.-S., Z.M.P., I.Y., A.Z., M.T.W., P.B., B.D., O.F., J.G., I.G., V.J.M., T.R.H., and I.V.K. Data curation, investigation, methodology: I.E.V., I.K., A.J., M.A., A.J.G., S.I., J.F.K.-S., K.U.L., S.E.P., R.R., A.W.H.Y., B.D., T.R.H., I.V.K., and M.T.W. Formal analysis, software, investigation, methodology: I.E.V., I.K., S.A., A.B., G.A., M.A., K.F., A.J.G., N.G., S.K., P.K., J.F.K.-S., V.N., Z.M.P., D.P., M.-L.P., I.Y., A.Z., P.B., O.F., J.G., and I.V.K. Project administration, funding acquisition, resources: B.D., I.G., F.A.K., V.J.M., T.R.H., and I.V.K. Supervision: P.B., I.G., F.A.K., J.G., B.D., J.F.K.-S., I.V.K., T.R.H., and V.J.M.

## Competing interests
O.F. is employed by Roche. All other authors declare no competing interests.

## Additional information

[1]Vavilov Institute of General Genetics, Russian Academy of Sciences, Moscow, Russia. [2]Institute of Protein Research, Russian Academy of Sciences, Pushchino, Russia. [3]Faculty of Bioengineering and Bioinformatics, Lomonosov Moscow State University, Moscow, Russia. [4]Altius Institute for Biomedical Sciences, Seattle, WA, USA. [5]Donnelly Centre and Department of Molecular Genetics, Toronto, ON, Canada. [6]École Polytechnique Fédérale de Lausanne, Lausanne, Switzerland. [7]Bioinformatics Competence Center, Ecole Polytechnique Fédérale de Lausanne, Lausanne, Switzerland. [8]Bioinformatics Competence Center, Université de Lausanne, Lausanne, Switzerland. [9]Institute of Organic Chemistry and Biochemistry of the Czech Academy of Sciences, 160 00 Praha 6, Staré Město, Czech Republic. [10]Computer Science Institute, Faculty of Mathematics and Physics, Charles University, 118 00 Praha 1, Staré Město, Czech Republic. [11]Laboratory of Systems Biology and Genetics, Institute of Bioengineering, School of Life Sciences, École Polytechnique Fédérale de Lausanne, Lausanne, Switzerland. [12]Swiss Institute of Bioinformatics,

Lausanne, Switzerland. [13]Life Improvement by Future Technologies (LIFT) Center, Moscow, Russia. [14]Chugai Pharmaceutical Co., Ltd, Tokyo, Japan. [15]Department of Computational Biology, Sirius University of Science and Technology, Sirius, Krasnodar region, Russia. [16]Max Planck Institute of Biochemistry, Planegg, Germany. [17]Institute of Computer Science, Martin Luther University Halle-Wittenberg, Halle, Germany. [18]Biosoft.Ru LLC, Novosibirsk, Russia. [19]Cincinnati Children's Hospital, Cincinnati, OH, USA. [20]Department of Medical Genetics, Centre for Molecular Medicine and Therapeutics, BC Children's Hospital Research Institute, University of British Columbia, Vancouver, BC, Canada. [21]Bioinformatics Laboratory, Federal Research Center for Information and Computational Technologies, Novosibirsk, Russia. [22]Moscow Center for Advanced Studies, Moscow, Russia. [23]Institute of Biochemistry and Genetics, Ufa Federal Research Centre of Russian Academy of Sciences, Ufa, Russia. [27]Present address: Cancer Research UK National Biomarker Centre, University of Manchester, Manchester, UK. [28]These authors contributed equally: Ilya E. Vorontsov, Ivan Kozin. ✉e-mail: seva.makeev@cruk.manchester.ac.uk; t.hughes@utoronto.ca; ivan.kulakovskiy@gmail.com

## The Codebook/GRECO-BIT Consortium

Philipp Bucher [12], Bart Deplancke [11,12], Oriol Fornes [20], Jan Grau [17], Ivo Grosse [17], Timothy R. Hughes [5]✉, Arttu Jolma [5], Fedor A. Kolpakov [15,21], Ivan V. Kulakovskiy [1,2,23]✉, Vsevolod J. Makeev [1,22,27]✉, Mihai Albu [5], Marjan Barazandeh [5], Alexander Brechalov [5], Zhenfeng Deng [5], Ali Fathi [5], Chun Hu [5], Samuel A. Lambert [5], Kaitlin U. Laverty [5], Zain M. Patel [5], Sara E. Pour [5], Rozita Razavi [5], Mikhail Salnikov [5], Ally W. H. Yang [5], Isaac Yellan [5], Hong Zheng [5], Georgy Meshcheryakov [2], Giovanna Ambrosini [6,7,8], Antoni J. Gralak [11,12], Sachi Inukai [14], Judith F. Kribelbauer-Swietek [11,12], Marie-Luise Plescher [17], Semyon Kolmykov [15], Ivan Yevshin [18], Nikita Gryzunov [2,13], Ivan Kozin [2,3,28], Mikhail Nikonov [3], Vladimir Nozdrin [3], Arsenii Zinkevich [3], Katerina Faltejskova [9,10], Pavel Kravchenko [16], Sergey Abramov [1,4], Alexandr Boytsov [1,4], Vasilii Kamenets [1,22], Dmitry Penzar [1], Anton Vlasov [24], Ilya E. Vorontsov [1,28], Aldo Hernandez-Corchado [25], Hamed S. Najafabadi [25], Quaid Morris [26], Xiaoting Chen [19] & Matthew T. Weirauch [19]

[24]TUD Dresden University of Technology, Center for Molecular and Cellular Bioengineering (CMCB), Biotechnologisches Zentrum (BIOTEC), Tatzberg 47/49, Dresden, Germany. [25]McGill University and Génome Québec Innovation Centre, 740 Dr. Penfield Avenue, Room 7202, Montréal, Québec, Canada. [26]Memorial Sloan Kettering Cancer Center, Rockefeller Research Laboratories, New York, NY, USA.

