## [Transparent Peer Review file · Communications Biology]

Cross-platform motif discovery and benchmarking to explore binding specificities of poorly studied human transcription factors

Corresponding Author: Professor Ivan Kulakovskiy

Version 0:

Reviewer comments:

Reviewer #1

(Remarks to the Author)

General comments

=====

This article presents a large and in-depth collaborative work led by the GRECO-BIT international consortium, to assess the performances of different motif discovery algorithms to produce binding models enabling to recognise sequences bound by a given transcription factor (TF). It describes an extensive analysis of the binding specificity for a wide panel of transcription factors, characterised by 5 different experimental methods ("platforms") : ChIP-seq, high-throughput SELEX, genomic HT-SELEX, protein binding microarrays and SMILE-seq. The authors developed a workflow that compares the motifs obtained from 10 alternative motif discovery algorithms, and evaluates the performances of each motif by its capacity to recognize sequences bound by a given transcription factor (TF). The evaluation is led in two configurations bringing complementary indications : (1) cross-validation among the sequences returned by one specific platform (train on a half dataset and test on the other half) ; (2) performance of the motifs discovered on one platform to identify the sequences from the other platforms. The results are made available in a motif catalog covering ~300 transcription factors, most of which had not been characterized so far.

Beyond the assessment of the TF binding models, an added value of the article is to present an original methodology to assess motifs produced with one experimental platform based on their capability to recognize TF binding sequences characterized by another method. The full accessibility of the MEX dataset (available on Zenodo) will be particularly valuable to evaluate performances of other approaches, e.g. alternative motif discovery algorithms, alternative parameters, motif-free machine learning approaches, etc.

Some pitfalls of the work are pointed out by the authors themselves in the discussion. In particular, the three top-performing motif discovery tools are those whose developers directly participated in the study, which clearly raises a bias, since these developers had the opportunity and incentive to fine-tune the parameters for their tools, which has not been done for the other tools. This is mentioned on lines 606 to 609, in the discussion. However, it would be important to highlight this much earlier in the manuscript, because the reader has to be aware it before interpreting the results, since all the software tools were not benchmarked in the same conditions.

Also, the authors highlight the difficulty in defining an ideal metric to quantify the performance of a motif or a motif-free approach, so they compute an average of different statistics. This raises a question about the robustness of the results to the somewhat arbitrary choice of a performance metric. Therefore, most results are presented in terms of ranks, which are not very helpful to understand the respective qualities of the motifs. In general, I am not sure that the average of several partly unsatisfying statistics would produce a better result than the careful use of a single well-understood and interpretable statistic (e.g. AUROC), or the comparison of a pair of statistics measuring the balanced performances (e.g. TPR vs FPR). Anyway, I recognize that this is not a simple question, so the choice of this average is understandable, but it would have been interesting, in parallel, to attempt understanding the properties that affect the performances of a motif, beyond reductionist parameters such as information content, matrix length).

Another valuable output of the work is the availability of all the discovered motifs on the CodeBook motif explorer website. For naive users, it might be confusing to be presented with several hundreds alternative motifs for each transcription factor, generally presenting strong similarity, but with some perceptible differences between them. However, for more advanced users it might be useful to dispose of a more complete collection than the top-ranking motif, which can in some cases be questionable (see Detailed Comment nb 1 below).

The last section of the results presents the evaluation of Random Forests to discriminate TF bound sequences from a negative data set (with 3 alternative ways of defining the negative set), based on the scores achieved by all the motifs discovered for this TF. Not surprisingly, the Random Forest performs better than any of the isolated motifs, but an interesting observation of the authors is that a combination of 2 to 4 PSSMs is generally sufficient to drastically improve the performances. Here as well, it would have been interesting to examine, at least for a few cases, how these 2 to 4 motifs compare with each other, and whether they bring in complementary information (e.g. motifs for the TF of interest + for other TF frequently bound in the same enhancers). Of course, this would not be feasible for each TF of the compendium of data, but it might be interesting to pick up a few study representative cases to highlight the link between some specific transcription factor, the “shape” of its motif (logos), and the performance statistics.

In summary, the consortium presents in this manuscript a consequent work with highly valuable data and results, and think that the work led by this consortium is a landmark in the domain, and is suitable for publication in Nature Comp Biol. The manuscript would benefit of some revisions to enhance its readability (some comments below) and include some elements of biological interpretation of the results.

Specific comments

=====

(1) An example for my concerns regarding the relevance of the scoring approach: the TF named AC092835 has 288 motifs in the Codebook, (<https://mex.autosome.org/approved/tf/AC092835>), each ranked according to two datasets (THC_0191 and THC_0199, respectively). Dimont ranks first on both scores, yet the examination of the motif logo shows a very “diluted” motif, with a poor information content. In contrast, the top-ranking motifs produced by other tools (ChipMunk, Streme, MEME) are very similar to each other, have an apparently informative logo, with the features expected from a C2H2 Zinc Finger DNA binding domain (each finger recognizes a oligonucleotide of length 3-4). In addition, for the other motifs, the ranks are quite contrasted between THC_0191 and THC_0199. I am skeptical about the relevance of the Dimont motif to reflect the TF binding properties, but for some reason the scoring scheme ranked this motif first.

(2) lines 64:66 : decision tree or random forest?

“By combining multiple PMWs in decision trees, we demonstrate how our setup can be readily adapted to train and test binding specificity models more complex than PWMs.”

In the abstract, why do you mention “decision trees” rather than Random Forest, which is the actual machine learning approach used in the last section.

(3) Lines 193-196: PSSM vs PWM

‘For clarity, in this study, we use the term “motif” to refer to a DNA binding specificity pattern, and “PWM” to refer to a motif model with one of two interchangeable representations: (1) a matrix of normalized nucleotide frequencies (also called the Position Frequency Matrix, PFM) or (2) a log-odds position weight matrix.’

This definition is a bit confusing. Firstly, using “pattern” to define “motif” does not bring any information, if you don’t define “pattern”. These terms were used more or less interchangeably in the domain of cis regulation, although motifs became predominant. I would suggest to find a more precise definition of the concept of motif (irrespective of the modes of representation, e.g. consensus, PSSM, HMM, ...).

More importantly, using “PWM” to refer to either PFM or PWM is really confusing. I would suggest to use Position-Specific Scoring Matrix for the general term, and to use PWM exclusively for the log-odds position weight matrix.

The term “normalized” is also a bit confusing. Do you simply mean “relative frequencies” (i.e. the sum of each matrix column is 1) or is there a more subtle normalization method (e.g. with pseudo-weights) ?

(4) line 201: “benchmarking protocol from 17 with additions from 10 and 37”. Not very readable at first sight. Please check with editors if this is the correct way to cite.

(5) Line 337: “we rescaled the values across motifs into the range [0,1], 0 corresponding to the worst and 1 to the best of values achieved by different motifs”

What is the interest of rescaling metrics? In particular, I find it questionable to rescale scores like AuROC or AuPRC, whose scale is intrinsically bound to [0,1]. The original scale should be conserved to keep the interpretability of these metrics. Note

that you highlight yourself the problem in the next paragraph (starting line 364), but in this case, is there any reason for not letting AuROC and AuPRC in their original scale.

(6) Figures 3C and 3D are hard to understand. The legend is unclear and provides insufficient detail to enable the reader to understand what is shown. The comment of this figure in the text (line 393) is even less informative. These figures apparently compare two coefficients of correlations. The meaning of these two correlations is not clear to me. I don't understand what the blue scale represents. The heat map shows that most values are concentrated around the diagonal.

(7) lines 393:39: "Further, the overall performance at the training and the test data types are highly correlated, suggesting that motif performance measurements with successful experiments from any platform are predictive regarding motif applicability to other data and that high-scoring motifs likely provide a good and generalized representation of the true binding specificity of a TF of interest."

This is really not clear to me. You discuss yourself about the potential impact of context-sensing pieces in some motifs, and when looking at the motifs, the top-ranked do not seem very informative (see my first remark in the section "detailed comments").

Reviewer #2

(Remarks to the Author)

N/A

Reviewer #3

(Remarks to the Author)

Summary:

Vorontsov and Kozin et al. describe an extensive cross-platform DNA motif discovery and benchmarking that involves multiple data types (from ChIP-seq to PBM) and various popular motif discovery tools. Furthermore, the authors are exploring TF-DNA interaction data for understudied TFs, making this study a significant contribution to the field. One of the key outcomes of this study is a publicly available resource (database) of motifs and related metadata. The scientific community studying transcription regulation will benefit significantly from the newly generated data, as well as from the extensive analysis and benchmarking.

The manuscript is well-structured and written. It includes all methodological details in addition to providing resources for reproducibility. The main findings are described and discussed well. We only have some minor comments on the manuscript and the database. We believe addressing them will improve the paper's readability and ease the database's navigation.

Comments:

1. What is the authors' motivation behind having two rounds of motif discovery? It is understood that the motifs from two rounds are pooled at the end. Did you check the redundancy between the first and second-round motifs? Why only a selection of tools was used for the second round?
2. In data pre-processing, input was prepared in several ways for different tools. For example, only the top 500 ChIP-seq peaks were used for discovery with STREME, ChIPMunk, and a few other tools, while for Dimont, all peaks were used. Can the authors comment on how much data preparation differences affect the motif discovery outcome?
3. Related to (1), different parameters were used for different motif discovery tools. Consider the number of motifs a tool can discover in a single dataset. You allow for various numbers and take different numbers of top motifs for further analysis (for example, top 3 motifs from STREME and top 5 from HOMER). Does this give an advantage to specific tools over others in your benchmarking? To better understand de novo motif discovery parameters and compare them, a supplementary table listing all parameter values will be helpful.
4. A selection of popular and widely used motif discovery tools were used. What exactly motivated this particular selection? Even with similar performances, some tools are better in some specific scenarios. In the end, you provide motifs discovered by all the different tools. But what are your overall recommendations for the community when performing motif discovery tasks? Should some tools be avoided when analyzing certain data types? Or some can be almost universally used?
5. Filtering of the artifacts improves the final set of motifs. But to what extent are the datasets affected by these artifacts? Based on your analysis, should this type of filtering be included in the discovery tools or applied by the users running routine motif discovery tasks?
6. STREME, Autoseed results were not discussed. ExplaiNN and ExplaiNN results were not discussed extensively.
7. Figure 1C, D radar plots are complex and challenging to get intuitively. A simple stacked barplot may be more straightforward.
8. Are the differences between methods in Figure 2 C and D significant (whatever the meaning used here)? The "*" is indicated but not explained in the figure legend.
9. Line 321: "First reported motifs scored higher...". Should it be a "motif"?
10. Common artifacts can be a valuable resource for users. At the moment, those motifs are provided as a .xlsx. It would be great to have those in a conventional cluster-buster, transfac, meme, or jaspas format.
11. Some plots in the supplementary section do not explain color scales—for example, Figures SF8A, B, D, and E.
12. Line 523: The abbreviation RF was not introduced earlier in the text.

13. Figure 5 should be moved to the Supplementary material as the first figure. It is never mentioned in the results section; therefore is out of context.

14. <https://github.com/autosome-ru/greco-bit-data-processing> was not accessible at the time of review.

15. The database/resource of motifs:

15.1. It would be great to provide more extensive documentation on the website itself. For example, describe what each column (experiment type, ranks, metrics) means. For instance, brief information about the columns could be indicated when hovering over it.

15.2. When switching to metrics, what is the threshold for a good metric? The values are colored (I think by the experiment type), but how should the user interpret the intensity of the color?

15.3. It is possible to sort individual rank columns for each technique. The rank for different approaches often does not match, and sometimes, the difference in ranking is huge. Would it be possible to filter the table and get motifs that ranked 1st overall and in each technique? It would be easier to compare such motifs visually.

15.4. TFBS prediction, especially improving TFBS prediction with a more advanced approach, is a significant contribution to the paper. Would it be helpful to provide those predictions to the users?

15.5. Having a search bar on individual TF pages could be nice so one does not have to return to the main page to search for the next TF.

Version 1:

Reviewer comments:

Reviewer #3

(Remarks to the Author)

We thank the authors for revising the manuscript and the web resource and providing extensive answers to all the reviewers' comments, which were well addressed. We have only one minor comment left.

In the abstract, you summarize that "..., suggesting that motifs with low information content can accurately represent the sequence specificity". We believe that this claim is too strong, as the performance of such motifs may depend on how the tests are constructed and on the specific test sets considered. For instance, it is unclear if such motifs would still perform well in genome-wide predictions. We suggest toning down the claim (for example, "..., motifs with low IC can perform well in our tested settings") or removing that particular part of the sentence.

Reviewers' comments

Reviewer #1 (Remarks to the Author)

General comments

=====

This article presents a large and in-depth collaborative work led by the GRECO-BIT international consortium, to assess the performances of different motif discovery algorithms to produce binding models enabling to recognise sequences bound by a given transcription factor (TF). It describes an extensive analysis of the binding specificity for a wide panel of transcription factors, characterised by 5 different experimental methods ("platforms") : ChIP-seq, high-throughput SELEX, genomic HT-SELEX, protein binding microarrays and SMILE-seq. The authors developed a workflow that compares the motifs obtained from 10 alternative motif discovery algorithms, and evaluates the performances of each motif by its capacity to recognize sequences bound by a given transcription factor (TF). The evaluation is led in two configurations bringing complementary indications : (1) cross-validation among the sequences returned by one specific platform (train on a half dataset and test on the other half) ; (2) performance of the motifs discovered on one platform to identify the sequences from the other platforms. The results are made available in a motif catalog covering ~300 transcription factors, most of which had not been characterized so far.

Beyond the assessment of the TF binding models, an added value of the article is to present an original methodology to assess motifs produced with one experimental platform based on their capability to recognize TF binding sequences characterized by another method. The full accessibility of the MEX dataset (available on Zenodo) will be particularly valuable to evaluate performances of other approaches, e.g. alternative motif discovery algorithms, alternative parameters, motif-free machine learning approaches, etc.

Some pitfalls of the work are pointed out by the authors themselves in the discussion. In particular, the three top-performing motif discovery tools are those whose developers directly participated in the study, which clearly raises a bias, since these developers had the opportunity and incentive to fine-tune the parameters for their tools, which has not been done for the other tools. This is mentioned on lines 606 to 609, in the discussion. However, it would be important to highlight this much earlier in the manuscript, because the reader has to be aware it before interpreting the results, since all the software tools were not benchmarked in the same conditions.

Indeed. We would like to emphasize that we did not aim to benchmark the tools or their alternative settings, but to compare the quality of the resulting motifs obtained from different experimental platforms with a unified strategy not requiring detailed parameter tuning. However, we tested alternative settings for classic tools, whose developers did not participate in the study, expanding their resulting motif repertoire, as described in Methods. We are aware

that the tools developed by the consortium members are likely to be in a privileged position, as the motif discovery parameters and preprocessing strategy could be better adapted to the available experimental data by the tools' authors. Of note, it did not help ExplainNN. **As requested by the reviewer, we have moved the respective statement to the first section of Results (Page 5, paragraph 3).**

Also, the authors highlight the difficulty in defining an ideal metric to quantify the performance of a motif or a motif-free approach, so they compute an average of different statistics. This raises a question about the robustness of the results to the somewhat arbitrary choice of a performance metric. Therefore, most results are presented in terms of ranks, which are not very helpful to understand the respective qualities of the motifs. In general, I am not sure that the average of several partly unsatisfying statistics would produce a better result than the careful use of a single well-understood and interpretable statistic (e.g. AUROC), or the comparison of a pair of statistics measuring the balanced performances (e.g. TPR vs FPR). Anyway, I recognize that this is not a simple question, so the choice of this average is understandable, but it would have been interesting, in parallel, to attempt understanding the properties that affect the performances of a motif, beyond reductionist parameters such as information content, matrix length).

While we fully agree that the single interpretable performance metric is desired, it does not seem realistic across different platforms and TFs. The ranges of observed performance are different for different TFs, platforms, and datasets, probably due to different compositions of binding sites and varying ranges of binding specificity. More to the point, it is unclear whether one should aim for a higher auROC or auPRC. Further, the absolute values of the classification performance (into binding vs. not-binding sequences) for TFBS models depend on technical features of the pipeline, as the method to generate negative sets (and class balance), and the "signal-to-noise ratio" of a particular experimental platform or a dataset. **We have added a note on this issue in 'Quantitative analysis of motif performance' (page 9, paragraph 4). Furthermore, the quantitative performance metrics can be explored in full on the MEX website.**

Also, in the end, we expect the user to arrive at a motif which is 'reasonably good' in different scenarios, and its comparatively good performance in multiple metrics helps in this regard. We explored the relationship between different metrics, and, while they do not agree perfectly, they are nonetheless highly correlated, meaning that the high-ranking motifs are reliably good in different scenarios. We also tried to explore the properties of a motif that could be predictive of its performance, but failed to find any simple way to measure the motif quality without running the actual benchmarks. **We have stressed this out in the revised version by changing the title of the respective section (page 13, paragraph 3) and adding a brief paragraph at the bottom (page 14, paragraph 2).**

Another valuable output of the work is the availability of all the discovered motifs on the CodeBook motif explorer website. For naive users, it might be confusing to be presented with several hundreds alternative motifs for each transcription factor, generally presenting strong similarity, but with some perceptible differences between them. However, for more advanced

users it might be useful to dispose of a more complete collection than the top-ranking motif, which can in some cases be questionable (see Detailed Comment nb 1 below).

We provide the top-20 motifs for download from the MEX website. By design, the MEX motif catalog includes motif subtypes for each of the tested TFs, although a detailed analysis of the subtypes requires a dedicated in-depth study. **In the context of the global benchmarking, we highlighted individual cases where distinct motif subtypes allowed for better TFBS prediction with Archipelago, see page 15, paragraph 3 in the revised manuscript and the newly added Supplementary Figure 9. The complete MEX motif set, along with the benchmarking results and rankings, is available for download from Zenodo, as stated in the 'Data Availability' section.**

The last section of the results presents the evaluation of Random Forests to discriminate TF bound sequences from a negative data set (with 3 alternative ways of defining the negative set), based on the scores achieved by all the motifs discovered for this TF. Not surprisingly, the Random Forest performs better than any of the isolated motifs, but an interesting observation of the authors is that a combination of 2 to 4 PSSMs is generally sufficient to drastically improve the performances. Here as well, it would have been interesting to examine, at least for a few cases, how these 2 to 4 motifs compare with each other, and whether they bring in complementary information (e.g. motifs for the TF of interest + for other TF frequently bound in the same enhancers). Of course, this would not be feasible for each TF of the compendium of data, but it might be interesting to pick up a few study representative cases to highlight the link between some specific transcription factor, the "shape" of its motif (logos), and the performance statistics.

While the cofactor motifs most certainly would allow improved predictions, in this study, we were specifically focused on binding specificities attributed directly to TFs of interest. In theory, cofactor motifs could emerge from genomic datasets, either ChIP-Seq or GHT-SELEX, due to composite elements found in the bound peaks. Yet, we selected only the top 20 motifs per ChIP-Seq or GHT-SELEX replicate to use in ArChIPelago. The inclusion of cofactor motifs in the feature set was further reduced by pre-filtering of common artifacts (see Methods, subsection "Filtering artifact signals") and including the peak centrality metric in ranking. With this in mind, as expected, we observed the highest gain from combinations of similar PWMs, corresponding to TF binding modes that are poorly described by a single PWM. **However, for some well-studied proteins with known alternative binding modes in vitro, such as FOSL2, Archipelago (trained either on ChIP-Seq or GHT-SELEX data) was not able to take them into account, and prioritized a distinct subtype for each of the two platforms. We have revised the respective paragraph in the manuscript, page 15, paragraph 3, and included the alternative motif logos for the selected TFs, achieving the highest gain with Archipelago in the newly introduced Supplementary Figure 9.**

In summary, the consortium presents in this manuscript a consequent work with highly valuable data and results, and think that the work led by this consortium is a landmark in the domain, and is suitable for publication in Nature Comp Biol. The manuscript would benefit of some

revisions to enhance its readability (some comments below) and include some elements of biological interpretation of the results.

Specific comments

=====

(1) An example for my concerns regarding the relevance of the scoring approach: the TF named AC092835 has 288 motifs in the Codebook, (<https://mex.autosome.org/approved/tf/AC092835>), each ranked according to two datasets (THC_0191 and THC_0199, respectively). Dimont ranks first on both scores, yet the examination of the motif logo shows a very "diluted" motif, with a poor information content. In contrast, the top-ranking motifs produced by other tools (ChipMunk, Streme, MEME) are very similar to each other, have an apparently informative logo, with the features expected from a C2H2 Zinc Finger DNA binding domain (each finger recognizes a oligonucleotide of length 3-4). In addition, for the other motifs, the ranks are quite contrasted between THC_0191 and THC_0199. I am skeptical about the relevance of the Dimont motif to reflect the TF binding properties, but for some reason the scoring scheme ranked this motif first.

First, the overall motif ranks between the datasets are in good agreement, as illustrated by the Kendall correlation of ~ 0.8 (check the 'Motif dataset similarity' heatmap on the respective MEX page:

Checking individual metrics, it becomes clear that the 'best hit'-based auROC metric fails to prioritize proper motifs in THC_0199, which might be related to technical features of an individual ChIP-Seq replicate such as peak length distribution. This is also related to the fact why we did not focus on a single performance measure as it can be non-informative or even misleading for particular experiments or platforms.

Second, the diluted Dimont motifs are often successful despite low information content. Many tools, such as MEME/ChipMunk/STREME, obtain the position frequencies directly from sequence alignment and the single 'best hits' from each sequence, and in these cases, the frequencies and information content reflect the composition of binding sites in the data. In contrast, Dimont directly optimizes the matrix values to discriminate TFBS-carrying sequences, it considers the whole spectrum of putative binding sites in each sequence, and does not rely on a direct sequence alignment. Thus, it is not surprising that Dimont motifs have lower information content. What is surprising, these motifs score highly in the consequent benchmarking. Of note, disregarding the IC, the top-ranking motifs from Dimont and ChipMunk are quite similar in terms of the core regions, where Dimont's motif is just a "flattened" version of ChipMunk's core motif region, subject to reverse complementary transformation.

Using the sequence logos to judge the motif reliability and predictive power has been a general approach in the field for decades, although it is prone to certain human biases [see e.g. Kok W., Oon Y.B., Lee N.K. Sequence logo visualization based on Gestalt perception (Novices vs experts) // 3rd International Conference of the South East Asian Network of Ergonomics Societies, SEANES2014. , 2014]. The overall information content is not only the most visually appealing attribute, but it could also be linked to biophysical properties of the binding specificity, and thus, we paid special attention to check how it relates to the motif performance (see the respective section on basic motif properties in Results).

First, if one considers only the 'best hit' in each sequence, Dimont (with its low-IC motifs) loses its edge (see page 21, paragraph 2 and Supplementary Figure 4), but remains highly competitive. Second, there was no correlation between a motif's predictive performance and its information content, for all performance metrics (see Supplementary Figure 6). All in all, we consider it an unexpected but highly important message to avoid judging the motifs by information content, and instead perform quantitative benchmarking whenever possible. Notably, we already reported this in Ambrosini *et al.* [see Figure S6 in doi:10.1186/s13059-020-01996-3] using a wide collection of previously available PWMs for well-studied TFs. We have added a note on this matter in the revised manuscript, see page 14, paragraph 2.

(2) lines 64:66 : decision tree or random forest?

"By combining multiple PMWs in decision trees, we demonstrate how our setup can be readily adapted to train and test binding specificity models more complex than PWMs. "

In the abstract, why to you mention "decision trees" rather than Random Forest, which is the actual machine learning approach used in the last section.

Indeed. Random Forest. FIXED.

(3) Lines 193-196: PSSM vs PWM

'For clarity, in this study, we use the term "motif" to refer to a DNA binding specificity pattern, and "PWM" to refer to a motif model with one of two interchangeable representations: (1) a matrix of normalized nucleotide frequencies (also called the Position Frequency Matrix, PFM) or (2) a log-odds position weight matrix.'

This definition is a bit confusing. Firstly, using "pattern" to define "motif" does not bring any information, if you don't define "pattern". These terms were used more or less interchangeably in the domain of cis regulation, although motifs became predominant. I would suggest to find a more precise definition of the concept of motif (irrespective of the modes of representation, e.g. consensus, PSSM, HMM, ...).

We tried to avoid the formal definition of the motif, which is given in formal language theory, but perhaps this is the only way to avoid confusion. **Thus, we have added a clarification statement in the respective paragraph (see page 4, paragraph 1).**

More importantly, using "PWM" to refer to either PFM or PWM is really confusing. I would suggest to use Position-Specific Scoring Matrix for the general term, and to use PWM exclusively for the log-odds position weight matrix.

We have revised the text (see page 5), clarifying the usage of PWM and PFM, following the reviewer's suggestion, although we do not use PSSM in general in the revised text to avoid an extra acronym.

The term "normalized" is also a bit confusing. Do you simply mean "relative frequencies" (i.e. the sum of each matrix column is 1) or is there a more subtle normalization method (e.g. with pseudo-weights) ?

Relative frequencies. **We have modified the text with a straightforward clarification ('sums to one'), see page 5.**

(4) line 201: "benchmarking protocol from 17 with additions from 10 and 37". Not very readable at first sight. Please check with editors if this is the correct way to cite.

We have modified the respective statement for better readability.

(5) Line 337: "we rescaled the values across motifs into the range [0,1], 0 corresponding to the worst and 1 to the best of values achieved by different motifs"

What is the interest of rescaling metrics? In particular, I find it questionable to rescale scores like AuROC or AuPRC, whose scale is intrinsically bound to [0,1]. The original scale should be conserved to keep the interpretability of these metrics. Note that you highlight yourself the problem in the next paragraph (starting line 364), but in this case, is there any reason for not letting AuROC and AuPRC in their original scale.

While it is tempting to keep the raw values and the scale of the benchmarking metrics, the effective scale differs dramatically depending on the experimental platform, particular dataset, or replicate, or benchmarking settings. For example, some over-enriched HT-SELEX datasets yield extremely high auROC values close to 1 (i.e., close-to-consensus binding sites are easy to distinguish). Also, sites in particular GHT-SELEX experiments are easier to predict (higher performance metrics across the board) compared to ChIP-Seq as the peaks do not depend on chromatin accessibility and reflect only direct binding (hence no contribution of cofactors). In the end, we normalized the metrics to [0,1] individually for each benchmark and each test dataset, making it possible to run a global comparison, although losing the direct

interpretability. We have highlighted this issue in the revised text, see page 10, paragraph 2, as also noted in our response to the previous comment above. Also, to partly counter this issue, we have also briefly explored raw values for individual metrics (see Figure 2, Supplementary Figure 4), but it did not yield any strong insights.

(6) Figures 3C and 3D are hard to understand. The legend is unclear and provides insufficient detail to enable the reader to understand what is shown. The comment of this figure in the text (line 393) is even less informative. These figures apparently compare two coefficients of correlations. The meaning of these two correlations is not clear to me. I don't understand what the blue scale represents. The heat map shows that most values are concentrated around the diagonal.

The Figures were intended to illustrate to what extent the measured performance on the train data type is predictive of the potential performance (normalized and averaged across metrics) at the target (different) data type, particularly, to highlight that the top-scoring motifs reliably transfer to non-seen data types, while lower-scoring motifs most likely fail to capture the binding specificity properly and do not recognize TFBS in the cross-platform setup. We wholeheartedly apologize for the confusing labels and the Figure caption. We have modified the respective Figure and the respective description in the text extensively and hope that the revised version provides sufficient details and properly delivers our message.

(7) lines 393:39: "Further, the overall performance at the training and the test data types are highly correlated, suggesting that motif performance measurements with successful experiments from any platform are predictive regarding motif applicability to other data and that high-scoring motifs likely provide a good and generalized representation of the true binding specificity of a TF of interest."

This is really not clear to me. You discuss yourself about the potential impact of context-sensing pieces in some motifs, and when looking at the motifs, the top-ranked do not seem very informative (see my first remark in the section "detailed comments").

This is related to the Figure above: based on cross-platform benchmarking, we conclude that the top-scoring (and top-ranked) motifs are the safe bet in terms of reflecting true binding specificity, while motifs with medium scores are often unreliable. Our original statement was not related to capturing the binding sites' context or to their information content, but to the benchmarking performance on the test data type (platform). We have modified the respective section in the revised manuscript, following the logic expressed in our previous comment. The motifs derived from synthetic data are, by definition, unable to account for the genomic context of binding sites (discussed in the section dedicated to the basic motif features).

Reviewer #2(Remarks to the Author):
Co-reviewer with Reviewer#1

Reviewer #3 (Remarks to the Author):

Summary:

Vorontsov and Kozin et al. describe an extensive cross-platform DNA motif discovery and benchmarking that involves multiple data types (from ChIP-seq to PBM) and various popular motif discovery tools. Furthermore, the authors are exploring TF-DNA interaction data for understudied TFs, making this study a significant contribution to the field. One of the key outcomes of this study is a publicly available resource (database) of motifs and related metadata. The scientific community studying transcription regulation will benefit significantly from the newly generated data, as well as from the extensive analysis and benchmarking.

The manuscript is well-structured and written. It includes all methodological details in addition to providing resources for reproducibility. The main findings are described and discussed well. We only have some minor comments on the manuscript and the database. We believe addressing them will improve the paper's readability and ease the database's navigation.

Comments:

1. What is the authors' motivation behind having two rounds of motif discovery? It is understood that the motifs from two rounds are pooled at the end. Did you check the redundancy between the first and second-round motifs? Why only a selection of tools was used for the second round?

Indeed, the whole collection contains many highly similar (and possibly redundant) motifs, often coming from the same round e.g., from software runs with alternative parameters, and we did not analyze the motif diversity in detail. The motivation for the two rounds of motif discovery was purely technical: we intended to reduce the computational cost by exploring the data with a small number of tools, curating the datasets based on the results, and running the 2nd extensive round of motif discovery on the approved subset of experiments only. Indeed, the whole analysis is based on the complete (pooled) collection of motifs, except for the dataset curation, which used only motifs obtained in the 1st round. **We have stressed this out in the revised text, see page 6, paragraph 4 in the first section of Results.**

2. In data pre-processing, input was prepared in several ways for different tools. For example, only the top 500 ChIP-seq peaks were used for discovery with STREME, ChIPMunk, and a few other tools, while for Dimont, all peaks were used. Can the authors comment on how much data preparation differences affect the motif discovery outcome?

The data preparation most likely affects the motif discovery results, but it is also linked to the software capabilities. For example, Dimont or ExplaiNN can efficiently handle large-scale datasets, while the MEME's computational cost increases quadratically, and thus only the top-scoring peaks are usually taken for motif discovery. We expect that using the larger datasets could improve the motif performance, but it remains practically unrealistic to run all the tools due to computational costs. In our setup, we tried to mimic the typical usage scenarios, where traditional motif discovery tools are used on a small subset of the most reliable sequences (top

peaks or pre-filtered HT-SELEX reads). We have included a note on this issue in the Discussion, see page 18, paragraph 3.

3. Related to (1), different parameters were used for different motif discovery tools. Consider the number of motifs a tool can discover in a single dataset. You allow for various numbers and take different numbers of top motifs for further analysis (for example, top 3 motifs from STREME and top 5 from HOMER). Does this give an advantage to specific tools over others in your benchmarking?

We did not try to balance the number of motifs per tool/approach (not only in terms of the number of reported motifs, but also in terms of the number of runs with alternative parameter combinations), and it indeed could handicap some tools at least due to multiple testing. However, as seen in Figures 1-3, the tools with the largest motif sets (MEME, STREME, and HOMER, see Supplementary Figure 1) did not show exceptional performance, while Dimont, with a medium-sized motif set, was a clear leader. We admit that our take on gkmSVM was of limited scale, and we performed a dedicated analysis of the gkmSVM motifs to ensure we do not misinterpret the findings (page 13). We have also found that, for ChIPMunk and Dimont, most of the time it could be enough to consider only the single top-ranked motif from the software output (see page 9, paragraph 2), as it was also the best scoring in the benchmarking.

To better understand de novo motif discovery parameters and compare them, a supplementary table listing all parameter values will be helpful.

Of note, we have included the key parameters for which we used non-default values for each of the tools in Methods. We consider it easier to keep these descriptions "as is", as parameter settings differ vastly between tools and platforms. We are ready to move these details from Methods to Supplementary materials, and would be glad to follow the Editors' advice on this matter.

4. A selection of popular and widely used motif discovery tools were used. What exactly motivated this particular selection?

The selection of tools was very straightforward: these were either widely popular tools (e.g. MEME, STREME, HOMER), or tools developed or routinely used by the consortium members (e.g. ChIPMunk, Dimont, Autoseed, ProBound).

Even with similar performances, some tools are better in some specific scenarios. In the end, you provide motifs discovered by all the different tools. But what are your overall recommendations for the community when performing motif discovery tasks? Should some tools be avoided when analyzing certain data types? Or some can be almost universally used?

The primary objective of our study was to compare different experimental platforms, particularly operating with synthetic and genomic DNA. Given the limitations of our setup discussed in the manuscript and mentioned above in this response document, we would like to avoid drawing too strong conclusions about the most and least successful motif discovery tools. While Dimont became a definitive leader across the board, other tools contributed significantly to the final collection, so the straightforward take-home message is to use less popular tools (such as Dimont) more often and for different platforms, and also perform motif discovery with a diverse toolbox, when possible. We mention the tools excelling in motif discovery for particular platforms in the "Benchmarking reveals versatility of individual approaches and added value of multiple tools" section of Results and in Discussion, and have **included the necessary conclusion in the revised text, see page 8.**

5. Filtering of the artifacts improves the final set of motifs. But to what extent are the datasets affected by these artifacts? Based on your analysis, should this type of filtering be included in the discovery tools or applied by the users running routine motif discovery tasks?

We consider the filtering of common artifacts to be an important feature of our pipeline, which should be included in any large-scale motif discovery analysis. Yet, we do not expect the particular set of motifs to be directly applicable to other studies, as the artifacts strongly depend on particular experimental protocols (e.g., a particular variant of HT-SELEX). Further, some common patterns which we partially excluded (like NFY- and ETS- motifs) might be meaningful and should not be discarded in the case of 'natural' ChIP-Seq performed w/o overexpression of a tagged protein. We discuss this issue **in the revised text, see pp. 18-19.**

6. STREME, Autoseed results were not discussed. ExplainNN and ExplainNN results were not discussed extensively.

Unfortunately, we did not gain any particular insights regarding these tools, although Autoseed was successful for HT-SELEX (page 10, paragraph 1), while ExplainNN was unsuccessful across the board (see page 11, paragraph 11). **We have added a note on STREME versus MEME in terms of the speed and accuracy trade-off, see pp. 17-18.**

7. Figure 1C, D radar plots are complex and challenging to get intuitively. A simple stacked barplot may be more straightforward.

We have replotted the panels as suggested.

8. Are the differences between methods in Figure 2 C and D significant (whatever the meaning used here)? The "*" is indicated but not explained in the figure legend.

We did not evaluate the pairwise differences between tools, but explored auROC as the raw performance metric by plotting the distribution of its values across the whole set of eligible <TF, dataset> pairs. In the revised Figure 2, we have removed the asterisks altogether.

9. Line 321: "First reported motifs scored higher...". Should it be a "motif"?

We initially had written this as 'motifs' as there were several tools, each with its first reported motif. We have changed it to 'motif' as suggested everywhere in the section.

10. Common artifacts can be a valuable resource for users. At the moment, those motifs are provided as a `.xlsx.` It would be great to have those in a conventional cluster-buster, transfac, meme, or jaspas format.

As suggested, we removed the xlsx table and instead provide the properly formatted files under 'Downloads' at the MEX website in MEME, JASPAR, HOMER, TRANSFAC, and HOCOMOCO formats. The same files are now also included in the primary MEX repository at Zenodo.

11. Some plots in the supplementary section do not explain color scales—for example, Figures SF8A, B, D, and E.

The respective 'missing' color codes were specified in the Figure captions, which were not included in the Supplementary Figures pdf of the original submission. In the revised version, following Communications Biol. policies, we have included the Figure captions along with the respective Figures.

12. Line 523: The abbreviation RF was not introduced earlier in the text.

FIXED.

13. Figure 5 should be moved to the Supplementary material as the first figure. It is never mentioned in the results section; therefore is out of context.

This Figure illustrates the procedure described in the Methods. We would like to keep it in Methods, given the journal policy permits it. **We would be glad to follow the Editor's advice on this matter.**

14. <https://github.com/autosome-ru/greco-bit-data-processing> was not accessible at the time of review.

We apologize for not ensuring that the repository was publicly accessible. **We have fixed the issue.**

15. The database/resource of motifs:

15.1. It would be great to provide more extensive documentation on the website itself. For example, describe what each column (experiment type, ranks, metrics) means. For instance, brief information about the columns could be indicated when hovering over it.

As the primary tables of the web interface are technically overloaded, we did not implement on-hover hints, **but updated the website with a simpler static Help section, reusing the materials previously included only in Supplementary materials.**

15.2. When switching to metrics, what is the threshold for a good metric? The values are colored (I think by the experiment type), but how should the user interpret the intensity of the color?

The color scale is independent and linear across values in each column. **We have included a note in the dedicated Help section.**

15.3. It is possible to sort individual rank columns for each technique. The rank for different approaches often does not match, and sometimes, the difference in ranking is huge. Would it be possible to filter the table and get motifs that ranked 1st overall and in each technique? It would be easier to compare such motifs visually.

We love the idea, but the currently implemented front-end is incapable of supporting this functionality without major refactoring. **Of note, the high-fidelity vector images of individual motif logos are downloadable directly from MEX ('save as image'). We now also provide archived logos for the overall top-ranking motifs in 'Downloads'.**

15.4. TFBS prediction, especially improving TFBS prediction with a more advanced approach, is a significant contribution to the paper. Would it be helpful to provide those predictions to the users?

Codebook ChIP-Seq and GHT-SELEX data are obtained from HEK cells with ectopic expression of the TFs of interest. While we consider these data to provide a great playground for demonstrating the model's capabilities, the genome-scale predictions do not seem to be widely useful in other contexts. **We now provide the trained models and the code suitable to make predictions in the ArChIPelago repository on GitHub and the accompanying Zenodo record. The data availability statement was updated accordingly.**

15.5. Having a search bar on individual TF pages could be nice so one does not have to return to the main page to search for the next TF.

The website backend does not include a search engine, **but we have included a TF selection menu on each page.**

Reviewer #3 (Remarks to the Author):

We thank the authors for revising the manuscript and the web resource and providing extensive answers to all the reviewers' comments, which were well addressed. We have only one minor comment left.

In the abstract, you summarize that "..., suggesting that motifs with low information content can accurately represent the sequence specificity". We believe that this claim is too strong, as the performance of such motifs may depend on how the tests are constructed and on the specific test sets considered. For instance, it is unclear if such motifs would still perform well in genome-wide predictions. We suggest toning down the claim (for example, "..., motifs with low IC can perform well in our tested settings") or removing that particular part of the sentence.

We have rephrased and toned down the sentence in the Abstract in the following way:

Notably, nucleotide composition and information content are not correlated with motif performance and do not help in detecting underperformers, while motifs with low information content, in many cases, describe well the binding specificity assessed across different experimental platforms.